# Enhanced elastocaloric cooling beyond Clausius–Clapeyron limits

Yuxin Song [1,2], Sheng Xu [1,2] ✉, Toshihiro Omori[2], Takuro Kawasaki [3], Yoshihisa Ishikawa [4], Ryoji Kiyanagi[3] & Ryosuke Kainuma[2]

The elastocaloric effect, driven by stress-induced martensitic transformations, offers a promising route toward efficient and environmentally friendly solid-state cooling. However, its practical implementation has been hindered by an inherent trade-off: materials exhibiting large isothermal entropy changes typically operate over narrow temperature windows, thereby limiting their overall cooling performance. Here, we demonstrate an elastocaloric response in a Ti–Al–Cr superelastic alloy that overcomes this limitation. Direct measurements reveal a pronounced elastocaloric cooling effect over an ultra-wide temperature range of 305 K, from 97 K to 402 K. This temperature span exceeds that predicted by the Clausius–Clapeyron relationship (235 K), indicating a significant deviation from conventional thermodynamic expectations. At room temperature, a large adiabatic temperature change of ~10 K is directly measured, corresponding to a cooling output of 5.76 J·g⁻¹ and a material coefficient of performance of 4.6, demonstrating competitive cooling performance at practical operating conditions. In addition, the elastocaloric response is maintained over the entire temperature range despite the expected decrease in entropy change at lower temperatures, indicating that the conventional trade-off between temperature span and cooling strength is effectively mitigated. This exceptional behavior originates from a combination of anomalous temperature dependence of the critical stress for martensitic transformation and high mechanical strength, which together enable fully reversible stress-induced transformations across a broad thermal domain. Our findings reveal a new regime of elastocaloric behavior and establish a guiding principle for overcoming the apparent limitations imposed by Clausius–Clapeyron-based descriptions in caloric materials.

Refrigeration systems, primarily based on vapor-compression technology, are essential to modern life but account for a significant portion of global energy consumption[1,2]. Despite their widespread adoption, conventional refrigeration technologies face critical limitations, particularly in terms of energy efficiency and their contribution to greenhouse gas emissions[3,4]. To address these challenges, solid-state cooling technologies, which are based on barocaloric, electro-caloric, magnetocaloric, and elastocaloric effects, have emerged as promising alternatives. These approaches rely on non-volatile solid materials with less global warming potential, offering the possibility to

[1]Frontier Research Institute for Interdisciplinary Sciences, Tohoku University, Sendai, Japan. [2]Department of Materials Science, Graduate School of Engineering, Tohoku University, Sendai, Japan. [3]J-PARC Center, Japan Atomic Energy Agency, Tokai, Japan. [4]Neutron Science and Technology Center, Comprehensive Research Organization for Science and Society, Tokai, Japan. ✉e-mail: xu.sheng.a8@tohoku.ac.jp

substantially reduce the carbon footprint of cooling systems[5–9]. Among these caloric effects, elastocaloric cooling, which exploits the latent heat associated with stress-induced martensitic transformations in superelastic alloys, has emerged as a leading candidate due to its potentially high coefficient of performance and scalable implementation[10–14].

A critical figure of merit for elastocaloric materials is their total cooling capacity, defined as the product (or integral) of entropy change ($\Delta S$) associated with martensitic transformation over a usable temperature span[15]. This quantity does not represent the practical refrigeration capacity of a single-stage cooling device, which is more appropriately evaluated as the integral of $\Delta S$ over a specific operating temperature at the cooling point, or equivalently as the product of the specific heat and the adiabatic temperature change. Instead, the total cooling capacity can serve as a descriptive thermodynamic metric to quantify the combined effect of a large entropy change and an unusually broad operational temperature window, which may guide the development of wide-temperature-range or cascaded elastocaloric cooling systems using a single material[12,16–18]. However, a fundamental thermodynamic trade-off limits this total cooling capacity: materials with large $\Delta S$ tend to exhibit strong temperature dependence of critical stress for inducing martensitic transformation, resulting in narrow operational temperature windows; conversely, materials with wide superelastic temperature ranges often possess small $\Delta S$, leading to limited cooling power. This trade-off is closely tied to the Clausius–Clapeyron relationship, which links $\Delta S$ to the temperature sensitivity of transformation stress[19,20]. As a result, most elastocaloric materials with a large cooling capability developed to date are confined to a narrow performance envelope, wherein improving one parameter inevitably sacrifices the other[21–24].

While breaking this trade-off has proven challenging according to the Clausius–Clapeyron relationship, a Ti–Al–Cr-based superelastic alloy we recently reported shows promising potential[25]. This alloy exhibits reversible stress-induced martensitic transformations from 4.2 K up to ~400 K in tension, while maintaining a moderately large, sign-consistent $\Delta S$ over a wide temperature range, although surprisingly the temperature dependence of critical stress changes from positive to negative below around 200 K. This observation challenges the prevailing assumption that the product of $\Delta S$ and temperature span is strictly constrained by the Clausius–Clapeyron-based thermodynamic framework. This prompted us to investigate the elastocaloric effect in detail. Here, we report an anomalous thermodynamic regime in a Ti–Al–Cr alloy, where the elastocaloric performance under compression exceeds Clausius–Clapeyron predictions, building upon but going significantly beyond our previous work[25]. We reveal a pronounced elastocaloric response with sustained cooling effect across an ultra-wide temperature range of 305 K, from 97 K to 402 K. The resulting total cooling capacity reaches 4726 J·kg$^{-1}$, exceeding the thermodynamic upper bounds predicted based on the Clausius–Clapeyron relation for temperature span and total cooling capacity by factors of 1.30 and 1.21, respectively.

## Results and Discussion
### Direct evaluation of elastocaloric effect over temperatures
We prepared high-quality single crystals of a Ti–Al–Cr alloy using a grain growth method[25]. Rectangular samples were cut and oriented for compression along the [001] direction, and mechanical tests were conducted across a wide temperature range from 4.2 K to 500 K. The stress–strain response reveals pronounced superelastic behavior across an exceptionally wide temperature range from 4.2 K to 460 K (Fig. 1a), with stress plateaus and full strain recovery, indicating reversible stress-induced martensitic transformation. Note that the superelasticity remains functionally stable with increasing loading–unloading cycles. Room-temperature cyclic tests reveal persistence of the hysteresis loop beyond 1000 cycles with only minor

residual strains (see *Supplementary Information*). The critical stresses associated with martensitic transformation, i.e., the martensitic transformation starting stress $\sigma_{Ms}$, the reverse transformation finishing stress $\sigma_{Af}$, and the equilibrium stress $\sigma_0$, as a function of temperature are summarized in Fig. 1b, where the definitions of $\sigma_{Ms}$ and $\sigma_{Af}$ are indicated in Fig. 1a and the $\sigma_0$ is approximated with $\sigma_0 = (\sigma_{Ms} + \sigma_{Af})/2$. Compared to our previous results under tension, which demonstrated superelasticity from 4.2 K to ~400 K, the current compression data indicate an even broader temperature range, likely due to improved resistance to fracture under compression[26]. Note that the temperature dependence of the $\sigma_0$ reverses sign at 150 K in compression, while a similar crossover occurs at ~200 K under tension[25]. The origin of this divergence between tensile and compressive behavior remains unclear and warrants further investigation. We also note that at cryogenic temperatures below 20 K, the compressive stress–strain curves exhibit pronounced serrated flow during both loading and unloading, being similar to that observed in cryogenic tensile tests[25]. These fluctuations are likely associated with dynamic pinning–depinning interactions between martensite/austenite interfaces and lattice defects such as dislocations and vacancies. Under such low thermal activation conditions, the interface mobility is strongly reduced, causing the transformation to proceed in an intermittent fashion. Despite these fluctuations, full strain recovery is retained, confirming the reversibility of the transformation. From an application standpoint, this behavior may be mitigated by improving lattice compatibility or reducing defect density through compositional tuning and thermomechanical processing. These strategies could help stabilize interface motion and suppress serration at low temperatures, further extending the utility of this alloy in cryogenic environments. Additionally, it is worth noting that in stress-driven martensitic transformation, strain compatibility across grain boundaries becomes a dominant factor, and intergranular constraint can limit reversible strain[27,28]. Therefore, configurations with reduced grain-boundary constraint, such as single-crystal samples, are more favorable for achieving large superelastic strain in the present alloy system.

Direct measurements of the elastocaloric effect were performed at selected temperatures across the superelastic range. Near-adiabatic temperature changes ($\Delta T_{ad}$) of the sample during rapid unloading were directly measured using a fast-response thermocouple in contact with the sample surface, following standard protocols for elastocaloric evaluation[21]. A distinct cooling effect (i.e., negative $\Delta T_{ad}$ upon unloading) was observed between 102 K and 402 K (Fig. 1c). The $\Delta T_{ad}$ reaches approximately −13.1 K at 402 K, with the magnitude of $\Delta T_{ad}$ gradually decreasing toward lower ends of the range. In contrast, below 102 K (i.e., 83 K, 65 K, 48 K), the sample is exothermic upon unloading, with $\Delta T_{ad}$ either vanishing or becoming slightly positive, suggesting that hysteresis-related heat dissipation dominates over the latent heat of transformation. At even lower temperatures below 48 K, $\Delta T_{ad}$ could not be directly measured due to technical limitations. Nevertheless, it is reasonable to assume that hysteresis-related energy dissipation and interfacial friction dominate in this regime. The serrated interface motion likely contributes to an even inverse (exothermic) $\Delta T_{ad}$ behavior during quasi-adiabatic unloading, where dissipative processes outweigh the latent heat effects. At temperatures above 402 K, $\Delta T_{ad}$ was not evaluated, as minor plastic deformation at high stress levels begins to affect the reversibility of the transformation and compromises reliable elastocaloric effect measurement. At room temperature (294 K), the material coefficient of performance (COP$_{mater}$), defined as the ratio of the cooling output per unit mass to the input work per unit mass (i.e., the hysteretic mechanical work), is calculated to be 4.6, based on a cooling output of 5.76 J·g$^{-1}$ derived from a $|\Delta T_{ad}|$ of 9.8 K. This value is comparable to that of commercial coarse-grained NiTi elastocaloric materials[7,29]. Together, these results demonstrate that the Ti–Al–Cr alloy exhibits a remarkably wide elastocaloric operating window over 300 K. The coexistence of wide-range

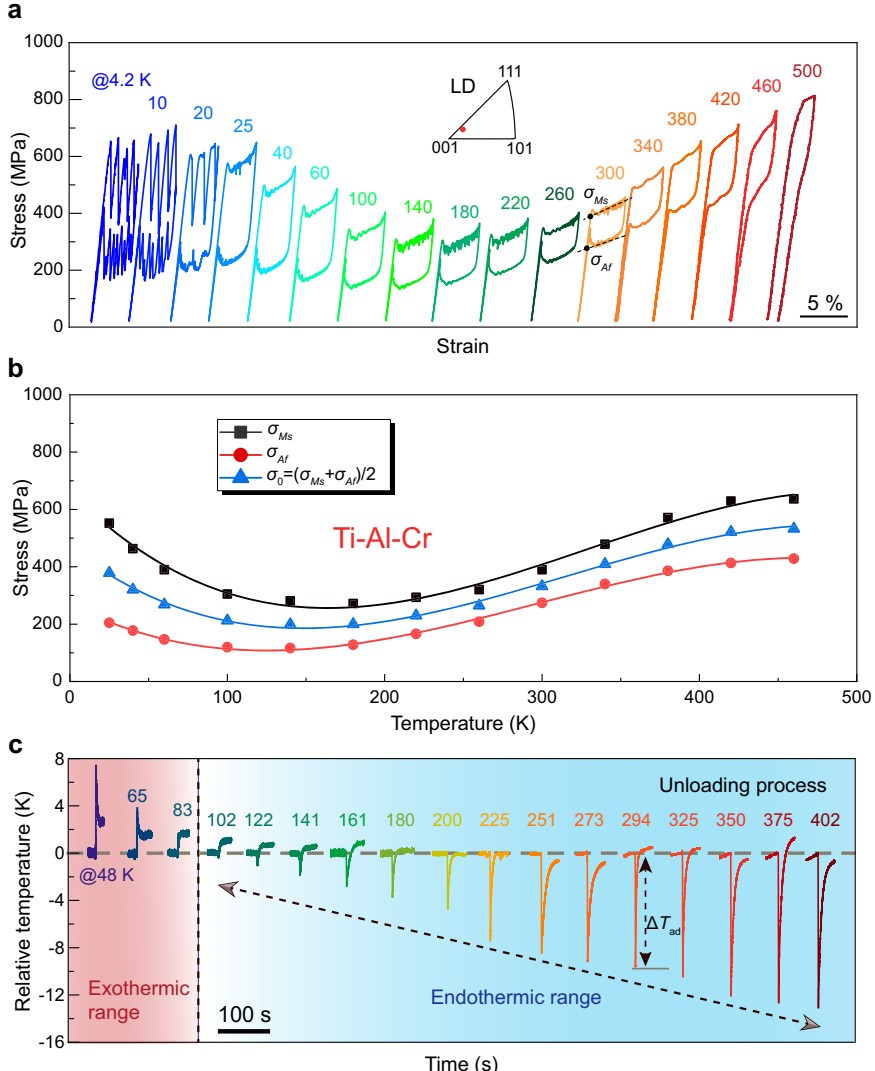

**Fig. 1 | Superelastic and elastocaloric properties of Ti–Al–Cr single crystal in compression at various temperatures. a** Stress–strain curves measured along the [001] direction from 4.2 K to 500 K, showing fully reversible superelasticity up to ~460 K. **b** Temperature dependence of superelastic behavior at various temperatures, showing the critical stresses: $\sigma_{Ms}$, $\sigma_{Af}$, $\sigma_{0}$ ( = ($\sigma_{Ms}$ + $\sigma_{Af}$)/2). **c** Relative adiabatic temperature change ($\Delta T_{ad}$) during unloading at various temperatures, determined from maximum temperature differences. A pronounced cooling effect is directly observed between 102 K and 402 K (endothermic range), while exothermic behavior dominates below 83 K.

superelasticity and sustained cooling response strongly indicates a rare combination of large transformation entropy and high phase stability over temperature—pointing toward the breakdown of the conventional $\Delta S$–usable temperature span trade-off.

## Structural evolution of phase transformation

To clarify the structural origin of the wide-temperature-range elastocaloric cooling behavior observed in the Ti–Al–Cr alloy, we performed in-situ neutron diffraction during uniaxial compression along the [001] direction at room temperature (Fig. 2a). Diffraction profiles collected under increasing stress reveal a direct transformation from the B2 structure of the parent phase to the B19 martensitic structure (Fig. 2b–e). Initially, only reflections from the cubic B2 phase—such as $001_{B2}$ and $110_{B2}$—are observed. With increasing stress, new peaks appear corresponding to $100_{B19}$ and $201_{B19}$, while the B2 peaks gradually disappear. This confirms a well-defined stress-induced phase transformation between two crystallographically distinct phases. Figure 2c plots the $d$-spacing strain along $[001]_{B2}$ orientation during the increasing compressive loading, which shows that the total

recovery strain consists of 2.3% elastic strain of the parent phase and 6.4% superelastic strain due to stress-induced phase transformation. The measured lattice parameter of the B2 phase under almost no stress is $a = 3.22$ Å. The martensite phase under high stress adopts an orthorhombic B19 structure with lattice parameters $a' = 2.94$ Å, $b' = 4.91$ Å, and $c' = 4.63$ Å. Diffraction intensity distributions in the reciprocal lattice space taken at low (40 MPa) and high (780 MPa) stress levels (Fig. 2d, 2e) further demonstrate that the transformation proceeds cleanly between the B2 and B19 phases. The orientation dependence of maximum recovery strain under uniaxial compression was plotted (Fig. 2f) using the determined lattice parameters and lattice deformation theory[30]. No other intermediate phases or diffuse scattering are detected throughout loading and unloading, indicating that the structural change is strictly two-phase and fully reversible. The absence of intermediate phases and the clear lattice correspondence between B2 and B19 provide a well-defined two-phase framework. This allows a reliable thermodynamic treatment of the transformation, particularly the estimation of entropy change based solely on the structural differences between the parent and

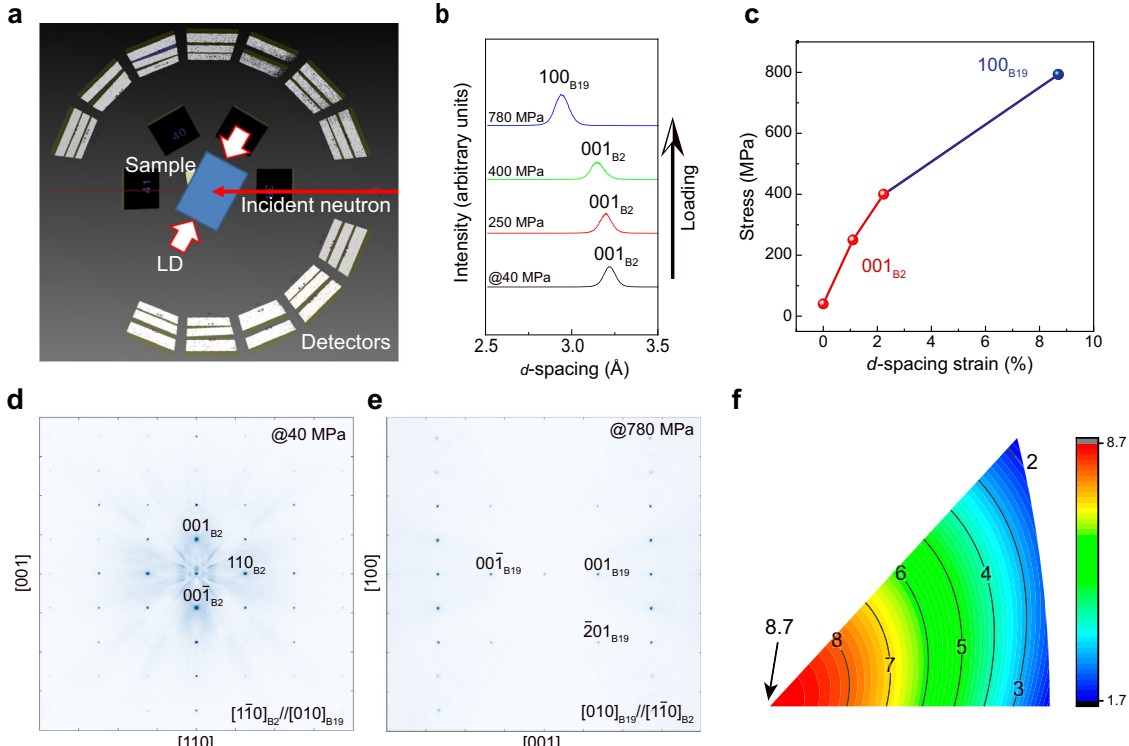

**Fig. 2 | Structural analysis during stress-induced martensitic transformation of a near-[001] oriented Ti–Al–Cr single crystal by a neutron diffractometer at room temperature. a** Schematic of the in-situ neutron diffraction measurement setup. LD, loading direction. **b** Variation in the $d$-spacing corresponding to the $(001)_{B2}$ crystal plane at various stress levels during loading, showing distinct phase transformation characteristics. **c** Corresponding $d$-spacing strain along the loading direction as a function of compressive stress. Neutron diffraction intensity distributions in the reciprocal lattice space were collected under loading stressed of (**d**) 40 MPa and (**e**) 780 MPa, showing B2 structure and B19 structure, respectively. **f** Calculated orientation dependence of recoverable strain under compression.

martensitic phases—forming the basis of the analysis presented in the following section.

## Thermodynamic analysis and predictions

To quantitatively assess the elastocaloric performance of the Ti–Al–Cr alloy, we carried out a thermodynamic analysis combining direct measurements with theoretical predictions. A central figure of merit for elastocaloric materials is the total cooling capacity, defined as the total amount of heat extractable via reversible martensitic transformation over a usable temperature range. Conventionally, this is estimated by integrating the isothermal entropy change ($\Delta S$) over temperature[31]:

$$Q_{cool} = \int_{T_1}^{T_2} \Delta S(T)\mathrm{d}T \qquad (1)$$

We previously determined $\Delta S(T)$ from calorimetric measurements of specific heat ($C_p$) for both B2 parent and B19 martensite phases[25]. As shown in Fig. 3a, the magnitude of isothermal entropy change increases with temperature, reaching −28.5 J·kg⁻¹·K⁻¹ at 300 K. Values beyond 300 K (red dots) were estimated using the Debye model (see *Supplementary Information*). This entropy change was derived from calorimetric measurements of separately stabilized parent and martensite phases and therefore reflects a stress-induced, rather than thermally induced, martensitic transformation. However, this conventional formulation presumes a fully reversible transformation, ignoring energy dissipation due to transformation hysteresis. In real materials, a portion of the mechanical work input is irreversibly dissipated as heat, thereby reducing the effective entropy change that contributes to cooling. To account for this, we introduce the effective entropy change:

$$\Delta S_{eff}(T) = \Delta S(T) - \frac{Q_{diss}(T)/2}{T} \qquad (2)$$

where $Q_{diss}/2$ is the dissipated energy derived from half of the area enclosed by stress–strain hysteresis loops, as presented in Fig. 3b. Accordingly, we redefine the effective total cooling capacity as:

$$Q_{eff} = \int_{T_1}^{T_2} \Delta S_{eff}(T)\mathrm{d}T \qquad (3)$$

To validate this approach, we compared the predicted adiabatic temperature change ($\Delta T_{ad}$) with direct experimental measurements using the following relation:

$$\Delta T_{ad} = -\frac{T \cdot \Delta S_{eff}}{C_p} \qquad (4)$$

Figure 3c compares the $\Delta T_{ad}$ (blue points) predicted from the calorimetrically determined $\Delta S$ (from $C_p$) with directly measured values (pink points). While the trends are consistent, the experimental $\Delta T_{ad}$ values are slightly smaller in magnitude. This discrepancy is likely attributed to imperfect adiabatic conditions and thermal losses during unloading, which reduce the actual temperature change. As such, the measured $\Delta T_{ad}$ values provide a realistic estimate of cooling performance under practical conditions, while the predictions based on Eq. (4) incorporating calorimetry-derived $\Delta S$ represent an upper limit. Deviations between calculated and measured adiabatic temperature

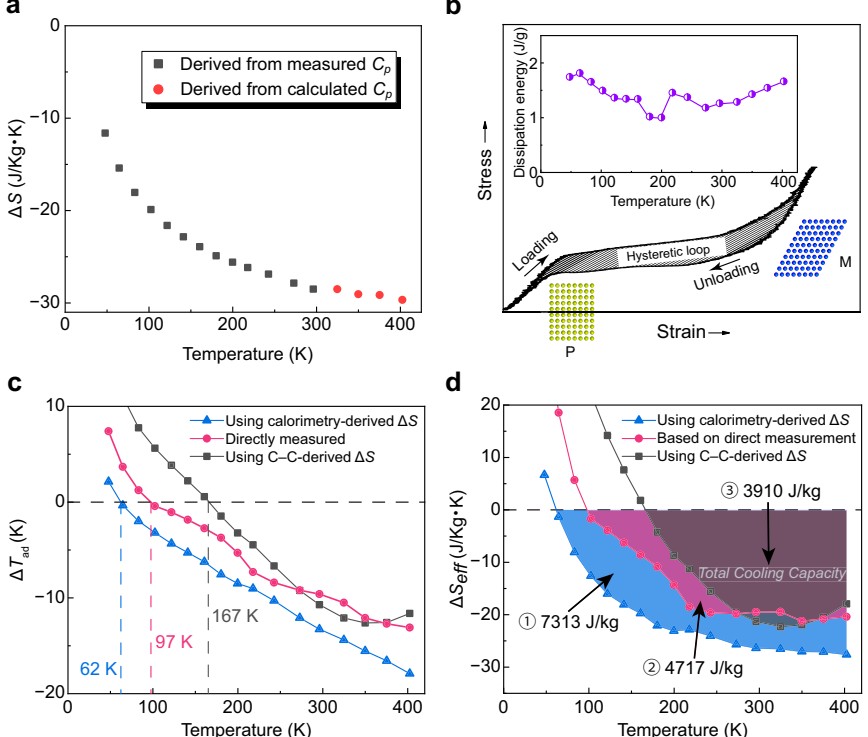

**Fig. 3 | Thermodynamic analysis of elastocaloric effect at various temperatures. a** $\Delta S$ of Ti–Al–Cr alloy determined by calorimetric measurements in previous work and thermodynamic estimations are plotted[25]. **b** Schematic illustration of dissipation energy caused by hysteretic superelastic loop, and the temperature dependence of dissipation energy. (**c**) Adiabatic temperature changes derived from Eqs. (2) and (4) using $\Delta S$ obtained from calorimetric measurements and experimentally measured $d\sigma_0/dT$, respectively, and experimentally measured $\Delta T_{ad}$, are shown. The lower temperature limit of elastocaloric cooling functionality is determined by the zero point of $\Delta T_{ad}$ in each line, respectively. **d** The temperature

dependence of effective entropy changes evaluated from the adiabatic temperature changes shown in Fig. 3c. The integration of the colored areas over the temperature range represents the effective total cooling capacity. The blue area corresponds to the effective total cooling capacity derived from calorimetric measurements. The pink area indicates the effective total cooling capacity obtained from experimentally measured $\Delta T_{ad}$. The dark area represents the effective total cooling capacity expected if the temperature dependence of superelasticity strictly followed the Clausius–Clapeyron relation.

changes primarily arise from imperfect adiabatic conditions during mechanical testing, sensor-related heat losses, and possible incomplete phase transformation under uniaxial compressive stresses. The experimentally observed $\Delta T_{ad}$ can also be used to inversely estimate $\Delta S_{eff}$.

The temperature dependence of $\Delta S$ can also be derived from the Clausius–Clapeyron relation in conventional shape memory alloys, which provides a more convenient approach as it does not require additional calorimetric measurements. This relation is typically expressed as[19]:

$$\frac{d\sigma_0}{dT} = -\frac{\Delta S}{\varepsilon} \tag{5}$$

where $\varepsilon$ is the transformation strain. The temperature dependence of $\Delta T_{ad}$ (dark points in Fig. 3c) was calculated using this relation, based on the temperature dependence of $d\sigma_0/dT$ in Fig. 1b, incorporating dissipation energy. According to this estimation, elastocaloric cooling functionality would disappear below 167 K, corresponding to the zero point of $\Delta T_{ad}$. However, direct experiments reveal that the Ti–Al–Cr alloy exhibits elastocaloric cooling effects down to 97 K, which is well below the predicted thermodynamic limit from the Clausius–Clapeyron relation. Furthermore, under ideal conditions, elastocaloric cooling functionality could persist to temperatures as low as 62 K, according to the $\Delta T_{ad}$ predicted from calorimetrically measured $\Delta S$ (i.e. from $Cp$). These discrepancies originate from the

suppressed $d\sigma_0/dT$ in the Ti–Al–Cr alloy, which fundamentally deviates from the temperature–stress scaling expected by the Clausius–Clapeyron relation (see *Supplementary Information*), thereby enabling a broader operational window for elastocaloric cooling. The suppressed $d\sigma_0/dT$ is attributed to the anomalous hardening of the B2 shear modulus at cryogenic temperatures, as previously reported (see also *Supplementary Information*)[25]. This behavior likely originates from increased lattice stiffness and reduced anharmonicity in the parent phase, possibly accompanied by local atomic ordering (e.g., ordered domain structures) or electronic effects. Although further investigation is needed, the absence of conventional microstructural hardening mechanisms (e.g., grain boundaries or precipitates) suggests that intrinsic lattice effects dominate the mechanical response at low temperatures. As a result, the operational window of 305 K (from 97 to 402 K) exceeds the 235 K span (from 167 to 402 K) predicted by the Clausius–Clapeyron relation by ~30%, yielding a total cooling capacity of 4726 J·kg⁻¹, a 21% enhancement over the conventional thermodynamic expectation of 3910 J·kg⁻¹ (Fig. 3d). Consequently, the Ti–Al–Cr alloy achieves a rare combination of large entropy change and broad thermal operability, surpassing the classical $\Delta S$–stress trade-off.

In particular, even within the low-to-intermediate temperature range of 100–220 K, where the transformation stress remains nearly constant (Fig. 1a, b), a pronounced elastocaloric cooling effect is still observed during unloading (Fig. 1c). In this regime, superelastic recovery and elastocaloric cooling functionality occur despite a nearly

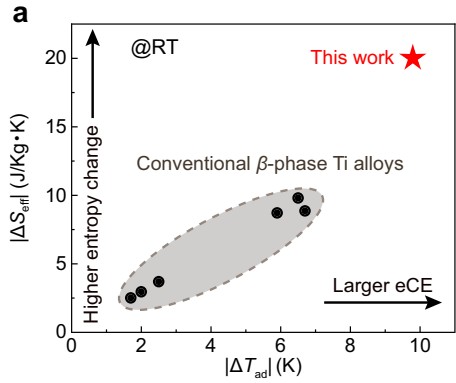
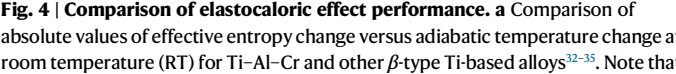
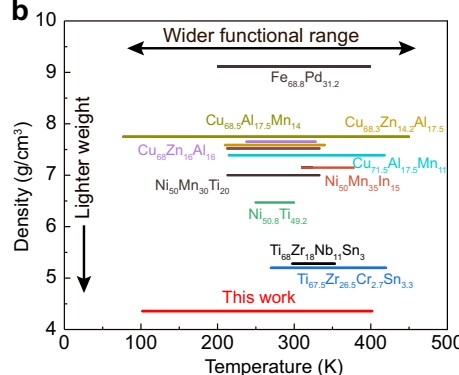

**Fig. 4 | Comparison of elastocaloric effect performance. a** Comparison of absolute values of effective entropy change versus adiabatic temperature change at room temperature (RT) for Ti–Al–Cr and other $\beta$-type Ti-based alloys[32–35]. Note that the $\Delta S_{eff}$ of each alloy was inversely estimated from experimentally observed $\Delta T_{ad}$. **b** Comparison of the working temperature range of elastocaloric effect versus density in the Ti-Al-Cr alloy with those in other elastocaloric alloys[21,31,36–41].

vanishing $d\sigma_0/dT$. Notably, the elastocaloric performance achieved in the current Ti–Al–Cr alloy still leaves substantial room for improvement compared to the intrinsic total cooling capacity of 7313 J·kg⁻¹ estimated from calorimetrically determined $\Delta S$ (from $Cp$). Nonetheless, these findings establish a new paradigm in elastocaloric material design, where anomalous thermo-mechanical responses, such as abnormal temperature dependence of shear modulus in a parent phase, can be exploited to realize cooling capacities beyond conventional thermodynamic limits estimated by the Clausius–Clapeyron relation.

It is important to distinguish between the effective total cooling capacity ($Q_{eff}$) and the specific cooling output ($Q_s$) used in evaluating COP$_{mater}$ (see *Supplementary Information*), which is expressed as:

$$Q_s = C_p |\Delta T_{ad}| \qquad (6)$$

Here, $Q_s$ describes the cooling energy produced per superelastic cycle under near-ambient conditions, whereas $Q_{eff}$ represents the total cooling energy that can be obtained over a given temperature span (e.g., in a cascading cooling configuration). While both quantities are useful, $Q_s$ provides a more common metric for evaluating elastocaloric performance in practical operation, where the temperature dependence of elastocaloric properties is often neglected.

While our previous report established the foundational discovery of cryogenic superelasticity under tensile loading in this alloy system, the present study substantially broadens the scope by demonstrating that similar functionality can be achieved under compressive loading[25]. This finding is particularly important because compressive configurations are more compatible with the requirements of practical elastocaloric cooling devices[12,26]. Moreover, we performed a detailed and quantitative comparison between the experimentally measured elastocaloric response and predictions derived from several thermodynamic approaches. This integrated framework uncovers a fundamental deviation with the conventional Clausius–Clapeyron relation, revealing that both the operating temperature range and the total cooling capacity significantly exceed certain theoretical expectations.

### Summary and implications

The Ti–Al–Cr superelastic alloy used in this study exhibits very high adiabatic temperature change ($\Delta T_{ad}$) and effective entropy change ($\Delta S_{eff}$) derived from experiment measurements among known Ti-based elastocaloric materials (Fig. 4a), marking a significant leap in elastocaloric performance[32–35]. Compared with conventional $\beta$-phase Ti alloys, which typically show $|\Delta T_{ad}|$ below 6 K and $|\Delta S_{eff}|$ near

10 J·kg⁻¹·K⁻¹, the present alloy achieves $|\Delta T_{ad}|$ of approximately 10 K and $|\Delta S_{eff}|$ approaching 20 J·kg⁻¹·K⁻¹ at room temperature. Moreover, thanks to the inherent low density of Ti alloys (-4.5 g·cm⁻³), this material is markedly lighter than traditional NiTi- or Cu-based shape memory alloys, which typically exceed 6 g·cm⁻³ [21,31,36–41]. To date, the largest adiabatic temperature change has been reported in the Ni–Mn–Ti system near room temperature owing to the strong coupling among martensitic transformation entropy, unit-cell volume change, and magnetic interactions, whereas the present Ti–Al–Cr alloy offers a much wider operating temperature range while maintaining a moderately large adiabatic temperature change[23]. As shown in Fig. 4b, it combines lightweight characteristics with an exceptionally wide operational window (97–402 K), making it uniquely suitable for aerospace applications where weight and broad-range cooling functionality are critical.

Beyond performance metrics, this work reveals a striking deviation from the classical Clausius–Clapeyron relation. While conventional wisdom holds that the temperature window of the elastocaloric effect is constrained by the $\Delta S$–stress coupling, our Ti–Al–Cr alloy maintains superelasticity and significant elastocaloric effect well below the theoretically predicted limit. This suggests that elastocaloric materials are not necessarily bound by traditional thermodynamic constraints based solely on Clausius–Clapeyron relation. Indeed, recent studies on nanostructured NiTi alloys also report similar anomalies[42,43], hinting at a broader principle: through rational alloy design, it may be possible to systematically suppress the temperature dependence of transformation stress, thereby unlocking new pathways for high-performance elastocaloric materials. Our findings thus not only establish a new benchmark for Ti-based cooling materials but also offer conceptual guidance for next-generation solid-state refrigeration technologies.

### Methods

A Ti$_{75.25}$Al$_{20}$Cr$_{4.75}$ alloy ingot was prepared by arc-melting pure elemental materials under an argon atmosphere. A block measuring 20 mm × 10 mm × 6 mm was first cut out from the ingot using electrical discharge machining (EDM) and subjected to cyclic heat treatment, consisting of repeated thermal cycling between 1200 °C and 700 °C, followed by quenching from 1200 °C to promote grain growth[25]. From this heat-treated block, single-crystal compression samples were prepared by EDM with the loading axis aligned along the desired crystallographic orientation, with a dimension of 5 mm × 2 mm × 2 mm. Prior to mechanical testing, all samples were mechanically ground and polished, and subsequently chemo-mechanically polished using a colloidal silica suspension. Mechanical compression tests were

conducted at a strain rate of $5 \times 10^{-4}\,\mathrm{s}^{-1}$ using a universal testing machine (Shimadzu Autograph AG-X 10kN) equipped with a contactless video extensometer for temperatures above the room temperature and a universal testing machine (Instron 5982) equipped with a customized cryo-chamber for temperatures below the room temperature. After stabilization of the superelastic response through mechanical cycling at room temperature (see *Supplementary Information*), a series of temperature-dependent loading–unloading tests were performed on a near [001]-oriented alloy single-crystal sample A across a wide temperature range from 4.2 K to 400 K. Based on the stress–strain behavior observed, the elastocaloric effect was then measured on a second single-crystal sample (sample B), which had the same orientation and underwent the same mechanical training process as sample A. The experimental protocol for elastocaloric effect measurement involved initial loading to predetermined stress levels at a strain rate of $1 \times 10^{-2}\,\mathrm{s}^{-1}$, followed by a holding period to achieve thermal equilibrium. Adiabatic temperature changes during rapid unloading (strain rate of $-0.2\,\mathrm{s}^{-1}$) were recorded using an E-type thermocouple welded to the sample surface. Supplementary Fig. 6 shows the temporal evolution of strain and temperature during loading–unloading mechanical testing at 294 K as a typical example. To ensure experimental safety and prevent strain overshooting, loading was performed at $1 \times 10^{-2}\,\mathrm{s}^{-1}$, while unloading was conducted at $-0.2\,\mathrm{s}^{-1}$ to approximate adiabatic conditions. We performed the in-situ neutron diffraction measurement during superelastic deformation of a near-[001]-oriented Ti–Al–Cr single crystal at room temperature, at a Time-of-Flight-Laue single crystal neutron diffractometer BL18 SENJU of J-PARC MLF. The experimental setup shown in Fig. 2a involved compressive loading of the single crystal while collecting neutron Laue diffraction intensities using wide-angle detectors.

## Data availability

The data that support the findings of this study are available from the corresponding author upon request.

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

## Acknowledgements

In situ neutron diffraction experiments at the Materials and Life Science Experimental Facility of the J-PARC were performed under a user program (No. 2022B0155). This work was supported by the Environmental Research and Technology Development Fund (Grant No. JPMEER-F20252RA2 to S.X.) of the Environmental Restoration and Conservation Agency provided by the Ministry of the Environment of Japan. This work was also supported by Grants-in-Aid for Scientific Research Fund (Grant Nos. 23K23070 to R. K., 21K18179 to T.O., 24K01190 and 24KK0264 to S.X., and 25K23519 to Y.S.) from the Japan Society for the Promotion of Science.

## Author contributions

S.X. conceived the project and designed the research. S.X. led the study and supervised Y.S., who performed sample preparation, microstructural characterization, mechanical testing, and in-situ neutron diffraction experiments, and analyzed the data. T.K., Y.I., and R. Kiyanagi assisted with the in-situ neutron diffraction measurements. Y.S., S.X., T.O., and R. Kainuma discussed and rationalized the results. Y.S. and S.X. wrote the original draft. S.X., T.O., and R. Kainuma reviewed and edited the manuscript. All authors discussed the results and approved the final version of the manuscript.

## Competing interests

The authors declare no competing interests.
