## [Transparent Peer Review file · Nature Communications]

Enhanced elastocaloric cooling beyond Clausius–Clapeyron limits

Corresponding Author: Dr Sheng Xu

Version 0:

Reviewer comments:

Reviewer #1

(Remarks to the Author)

The Authors identified an extremely wide temperature window for elastocaloric effect in a recently-developed Ti-Al-Cr alloy. Moreover, the behavior predicted by Clausius Clapeyron law has been overcome by the alloy, which has an inversion of the temperature trend of critical stress behavior and a superior mechanical strength, opening a perspective towards an innovative approach to design and develop new caloric alloys. In my opinion, this manuscript represents a cutting-edge study in the field of elastocaloric materials and can be considered for publishing in Nature Communications, but some minor revisions and modifications are still necessary before the acceptance. Below my observations:

- The Authors should specify which are the exact dimensions of the specimens cut from single crystal and which method was used for cutting.
- Extremely large fluctuations are shown in superelastic stress-strain curves at 4.2, 10 and 20 K, both in loading and unloading stage. Please comment this behavior that could actually prevent the practical application of this alloy at such low temperatures in space or harsh environments. Is this behavior affecting the increase of the inverse (exothermic) behavior during quasi-adiabatic unloading at low temperatures?
- In the graph reported in Figure 1.c, relative temperature axis should be corrected with positive values above 0.
- The main finding of the manuscript is the unexpected trend of critical stress with temperature, which leads to the unexpected superelasticity and elastocaloric effect at low temperatures. However, the causes of this anomalous behavior are not explained and described in deep by the Authors. Further discussion about the origin of the anomalous hardening of the shear modulus in the B2 parent phase should be provided, also considering possible microstructural aspects.
- Have the Authors considered the evolution of the anomalous mechanical behavior and thermal response upon mechanical cycling? Since the elastocaloric devices must work for several repeated cycles, the Authors should include some considerations about fatigue behavior and mechanical properties evolution of the studied alloy.

Reviewer #2

(Remarks to the Author)

Dear Editor, Dear Authors;

The manuscript presents an elastocaloric effect of novel shape memory Ti-Al-Cr alloy, which shows reversed superelasticity and elastocaloric effect in a very wide temperature range – from ultra-low towards room temperatures. This makes this alloy very interesting for cryogenic and room temperature applications. The paper is interesting and well written, but some comments should be addressed.

1. The authors claim a very high cooling capacity, which was calculated by multiplying isothermal entropy change and an available temperature range. I am aware that this metric has been used in literature before, but as such does not have any physical meaning according to my knowledge. The actual (meaningful) specific cooling capacity which elastocaloric material could produce should be calculated as a product of specific heat and the adiabatic temperature change or as a product of isothermal entropy change and a temperature (at which the cooling power is absorbed and not the entire possible temperature range).
2. Was that the same sample that was tested from 4.2 K to 400 K or was it a different sample for each temperature?
3. What was the strain rate at isothermal and adiabatic conditions? How the authors know that the selected strain-rate fulfills adiabatic conditions? Can they report adiabatic stress-strain responses?

4. On Fig. 3c the authors are comparing adiabatic temperature changes calculated/measured by different methods, which is very interesting. However, they include both temperature-induced (calorimetry) and stress-induced. The author should comment on that and explain the reasons for deviations more in detail.
5. I would suggest that the authors calculate the material COP: $\text{mass} \times \text{specific_heat} \times \text{adiabatic_temp_change}$ vs adiabatic hysteresis input work.

Reviewer #3

(Remarks to the Author)

In this manuscript (NCOMMS-25-83317-T), Song. et.al., reported an exceptionally large elastocaloric effect, in Ti-Al-Cr superelastic alloy, over an ultra-wide temperature range of 305 K with a cooling capacity reaching 4726 J/kg that is beyond Clausius-Clapeyron limit. They further explained their experimental observation originates from reversible stress-induced martensitic transition. In situ neutron diffraction measurements have been used to evaluate structural transition, during compression at room temperature. The manuscript is well written, but the following concerns must be addressed before considering in the "Nature Communications" journal.

1. The authors reported in their previous paper {Ref.22} {stress vs temperatures (Fig.3(a)), Supply Fig.9 (a)-9(b)} are similar to the Fig. 1(a) and (b) of the present manuscript. They claimed an ultra-wide temperature window, while they have already reported on their earlier work based on the Figures of Ref: 22}. What is the novelty of the present work? Authors need to explain in detail.
2. It is stated in the first paragraph of page 2 that this material exhibits reversible stress-induced martensitic transformations from 4.2 K up to ~400 K in tension. How do they confirm that martensitic transformation (MT) (from Cubic B2 to orthorhombic B19 transition) is reversible, as martensitic transition is always associated with strain and hysteresis? Because hysteresis leads to work loss in every heat pumping cycle as dissipated heat. Henceforth, the authors are instructed to provide compatibility relations along with the cofactor conditions with the austenite and martensite interface. It is also suggested to show microstructure morphology during transition. See the reference; Cui et. al., Nature Materials 5, 286–290 (2006).
3. It is requested to perform DSC measurement to determine the magnitude of hysteresis during transformation.
4. In general, the microstructure of the caloric materials during transformation changes drastically in consecutive transformation cycles, whereas macroscopic properties such as transformation temperature and latent heat are nearly reproducible. It is also suggested to show microstructure morphology during transition. See the reference Song et.al., Nature 502, 85–88 (2013).
5. The author should add material's efficiency of the present sample i.e., material's coefficient of performance (COP) to Carnot efficiency (COP) and a comparison figure for the material with other elastocaloric materials.
6. Have the authors checked their mechanical stability, such as bulk modulus B to shear modulus ratio and the Cauchy pressure of the material?
7. They performed the orientation dependence of recoverable strain under compression, which has depicted in Fig.2(f). In addition, a change of grain boundaries can enhance strain recoverability and elevate the utilization of latent heat during stress-driven MT. While improving strength by reducing grain size is an established physical principle, i.e., the Hall-Patch equation. I think it would be more persuasive and powerful if the author could comment on this and explain how the grain size of their sample can help improve strain recoverability?
8. In the Summary section, the authors compared β -phase Ti alloy with their findings. Recently, a colossal elastocaloric effect has been observed in a promising all-d-Heusler Ni-Mn-Ti ferroelastic system. Authors should add the ferroelastic Ni-Mn-Ti material in Fig. 4(a) and explain their findings with these systems. See the reference: [Cong. et. al., Physical Review Letters 122, 255703 (2019)].

Version 1:

Reviewer comments:

Reviewer #1

(Remarks to the Author)

The Authors properly answered to all the issues and queries. Moreover, they added substantial observations in the text and in some cases they provided additional tests to improve their discussion. Therefore, the quality of the manuscript is substantially improved and the relevance of the work is confirmed. In my opinion, the manuscript is now suitable for publication in the present form.

Reviewer #2

(Remarks to the Author)

Dear Editor; Dear Authors

The authors have successfully addressed most of my previous comments, however, one important issue remains. Although the authors agreed that the reported total cooling capacity does not represent the practical refrigeration capacity of a cooling device (and is thus as such more or less meaningless), they have still reported this value (in abstract, which might be misleading!). Later in the text (within the COP calculation) they have calculated also more representative cooling capacity (specific heat multiplied by adiabatic temperature changes), and I would strongly suggest reporting this value instead.

Version 2:

Reviewer comments:

Reviewer #2

(Remarks to the Author)

Dear Authors;

The revised manuscript now indeed reports the specific output cooling energy, which is much more relevant. However, I would strongly suggest updating the "Thermodynamic analysis and predictions" by adding the equation based on which the specific output cooling energy was calculated and comment on the difference between it and total cooling capacity (eq. (1)), which is still mentioned as a central figure of merit for elastocaloric material.

Version 3:

Reviewer comments:

Reviewer #2

(Remarks to the Author)

Thank you for the corrections. The paper can now be accepted as it is.

RESPONSE TO REVIEWERS' COMMENTS

We would like to express our sincere gratitude to the expert reviewers for their insightful comments and valuable suggestions. We have provided a point-by-point response to each comment, with the revised sections highlighted in the manuscript. These revisions have significantly enhanced the quality of our work, and we deeply appreciate their contributions.

Reviewer #1 (Remarks to the Author):

The Authors identified an extremely wide temperature window for elastocaloric effect in a recently-developed Ti-Al-Cr alloy. Moreover, the behavior predicted by Clausius-Clapeyron law has been overcome by the alloy, which has an inversion of the temperature trend of critical stress behavior and a superior mechanical strength, opening a perspective towards an innovative approach to design and developing new caloric alloys. In my opinion, this manuscript represents a cutting-edge study in the field of elastocaloric materials and can be considered for publishing in Nature Communications, but some minor revisions and modifications are still necessary before the acceptance. Below my observations:

Response:

We sincerely thank the reviewer for thorough effort in reviewing our manuscript and for the insightful comments. We are encouraged by the recognition of our original findings and their potential impact on the advancement of new elastocaloric materials.

1. - The Authors should specify which are the exact dimensions of the specimens cut from single crystal and which method was used for cutting.

Response:

We apologize for the omission of the specimen dimensions and cutting methods in the original manuscript.

We have now included the relevant details in the Methods section on Page 9 Line 32 in the revised manuscript, as follows:

“A $Ti_{75.25}Al_{20}Cr_{4.75}$ alloy ingot was prepared by arc-melting pure elemental materials under an argon atmosphere. A block measuring $20\text{ mm} \times 10\text{ mm} \times 6\text{ mm}$ was first cut out from the ingot using electrical discharge machining (EDM) and subjected to the cyclic heat treatment described in our previous report to promote grain growth^[25]. From this heat-treated block, single-crystal compression specimens were prepared by EDM with the loading axis aligned along the desired crystallographic orientation, with a dimension of $5\text{ mm} \times 2\text{ mm} \times 2\text{ mm}$. Prior to mechanical testing, all specimens were mechanically ground and polished, and subsequently chemo-mechanically polished using a colloidal silica suspension.”

2. - Extremely large fluctuations are shown in superelastic stress-strain curves at 4.2, 10 and 20 K, both in loading and unloading stages. Please comment on this behavior that could actually prevent the practical application of this alloy at such low temperatures in space or harsh environments. Is this behavior affecting the increase of the inverse (exothermic) behavior during quasi-adiabatic unloading at low temperatures?

Response:

We thank the reviewer for pointing out the pronounced stress fluctuations observed at 4.2 K, 10 K, and 20 K. These serrated features are likely attributed to dynamic pinning–depinning interactions between martensite/austenite interfaces and lattice defects such as dislocations and vacancies. Under cryogenic conditions, where thermal activation is minimal, the mobility of martensite/austenite interfaces is substantially suppressed. As a result, the stress-induced martensitic transformation proceeds in an intermittent, jerky manner, leading to abrupt stress drops during both loading and unloading. Despite these fluctuations, full macroscopic strain recovery was consistently achieved, indicating that the transformation remains reversible at the bulk level. However, we acknowledge that such serrated stress responses could potentially hinder precise actuation or stable energy conversion in practical cryogenic applications, such as aerospace or deep-space mechanisms, where mechanical smoothness is critical. We believe that these fluctuations can be mitigated by improving lattice compatibility through compositional tuning or orientation control, which may enhance interface mobility and reduce defect-induced pinning. This is an important direction for future research.

Regarding elastocaloric performance, we note that the quasi-adiabatic temperature change (ΔT_{ad}) is expected even to reverse (i.e., become exothermic) at these low temperatures during unloading. Although direct ΔT_{ad} measurements were not possible below 48 K due to experimental limitations in thermocouple attachment, this trend is consistent with our expectations. The serrated interface motion likely increases internal friction and dissipative interfacial processes, which can outweigh the latent heat associated with reverse transformation, resulting in net heat generation rather than absorption. Importantly, in this study, effective elastocaloric cooling was achieved above 100 K, where stress fluctuations are minimal. Therefore, the observed serrations do not compromise the practical elastocaloric cooling performance reported here.

We have added discussions related to these serration behaviors in the revised manuscript, on page 3 Line 10:

“We also note that at cryogenic temperatures below 20 K, the compressive stress–strain curves exhibit pronounced serrated flow during both loading and unloading, being similar to that observed in cryogenic tensile tests^[25]. These fluctuations are likely associated with dynamic pinning–depinning interactions between martensite/austenite interfaces and lattice defects such

as dislocations and vacancies. Under such low thermal activation conditions, the interface mobility is strongly reduced, causing the transformation to proceed in an intermittent fashion. Despite these fluctuations, full strain recovery is retained, confirming the reversibility of the transformation. From an application standpoint, this behavior may be mitigated by improving lattice compatibility or reducing defect density through compositional tuning and thermo-mechanical processing. These strategies could help stabilize interface motion and suppress serration at low temperatures, further extending the utility of this alloy in cryogenic environments.”

And on page 3 Line 32:

“At even lower temperatures below 48 K, ΔT_{ad} could not be directly measured due to technical limitations. Nevertheless, it is reasonable to assume that hysteresis-related energy dissipation and interfacial friction dominate in this regime. The serrated interface motion likely contributes to an even inverse (exothermic) ΔT_{ad} behavior during quasi-adiabatic unloading, where dissipative processes outweigh the latent heat effects.”

3. - In the graph reported in Figure 1.c, relative temperature axis should be corrected with positive values above 0.

Response:

We apologize for the wrong values at exothermic temperature ranges, and Figure 1c has been revised accordingly in the revised manuscript.

4. - The main finding of the manuscript is the unexpected trend of critical stress with temperature, which leads to the unexpected superelasticity and elastocaloric effect at low temperatures. However, the causes of this anomalous behavior are not explained and described in deep by the Authors. Further discussion about the origin of the anomalous hardening of the shear modulus in the B2 parent phase should be provided, also considering possible microstructural aspects.

Response:

We thank the reviewer for raising this important point. We agree that the anomalous temperature dependence of the critical stress is central to the unusually wide superelastic and elastocaloric operating window observed in this alloy, and that further discussion on its physical origin is necessary. It is worth noting that an increase in elastic moduli upon cooling is not, by itself, unusual for many B2 or bcc metals, where reduced thermal vibrations naturally lead to lattice stiffening. However, in the context of shape memory alloys, the more commonly observed behavior is elastic softening upon cooling toward the martensitic transformation, often associated with phonon softening or instability of specific shear modes. In this sense, the pronounced hardening of the shear modulus or Young’s modulus observed in the B2 parent phase of the present Ti–Al–Cr alloy at lower temperature represents a non-typical behavior for superelastic alloys. In

our previous work [Ref. 25], we experimentally identified this low-temperature elastic hardening behavior, but its implications for elastocaloric performance were not addressed. In the present manuscript, we extend this understanding by explicitly linking the shear modulus hardening to a suppressed temperature dependence of the equilibrium transformation stress, $d\sigma_0/dT$. This suppression directly weakens the constraint imposed by the Clausius–Clapeyron relation and enables both superelasticity and a measurable elastocaloric effect to persist to much lower temperatures than conventionally expected.

Regarding the microscopic origin of this hardening behavior, we note that it likely reflects a combination of intrinsic lattice effects and microstructural stability of the B2 phase. The absence of pronounced pre-transitional softening suggests that the parent phase is elastically stabilized at low temperatures, possibly due to reduced phonon anharmonicity, strong directional bonding, and the suppression of lattice instabilities. From a microstructural standpoint, the specimens in this study are essentially free of grain boundaries, and no visible precipitation was observed under SEM or TEM, ruling out conventional grain or precipitate hardening. However, further work using inelastic neutron scattering could help clarify the role of phonon behavior in this anomalous modulus evolution.

We have added this discussion to the revised manuscript on page 5 Line 32 as follows:

“The suppressed $d\sigma_0/dT$ is attributed to the anomalous hardening of the B2 shear modulus at cryogenic temperatures, as previously reported (see also Supplementary Information)^[25]. This behavior likely originates from increased lattice stiffness and reduced anharmonicity in the parent phase, possibly accompanied by local atomic ordering (e.g., ordered domain structures) or electronic effects. Although further investigation is needed, the absence of conventional microstructural hardening mechanisms (e.g., grain boundaries or precipitates) suggests that intrinsic lattice effects dominate the mechanical response at low temperatures.”

5. - Have the Authors considered the evolution of the anomalous mechanical behavior and thermal response upon mechanical cycling? Since the elastocaloric devices must work for several repeated cycles, the Authors should include some considerations about fatigue behavior and mechanical properties evolution of the studied alloy.

Response:

We thank the reviewer for highlighting the importance of mechanical cycling behavior for elastocaloric applications. To address this, we have performed additional compressive loading–unloading tests up to 1600 cycles at room temperature (see Supplementary Fig. 3, also attached below). The results demonstrate that the material retains a well-defined and reproducible stress–strain hysteresis loop, with a relatively large recoverable strain of approximately 4% after the

cycling tests. This indicates that the stress-induced martensitic transformation remains active and reversible even after extended cycling. Importantly, the persistence of the hysteresis loop and stable recoverable strain suggest that the dissipated energy remains relatively unchanged throughout the cycling process. This also implies that the entropy change associated with the transformation continues to be effective. Although gradual degradation of the adiabatic temperature change (ΔT_{ad}) is often reported in many elastocaloric materials, our results suggest that such degradation, if any, would likely be progressive rather than abrupt in the present alloy. When compared to lightweight disordered Ti-based or Mg-based SMAs (J. Ma et al, Acta Mater. 58, 2216 (2010); Y. Ogawa et al, Science 353, 368 (2016)), the Ti–Al–Cr alloy exhibits superior cyclic stability. Its performance is also comparable to, or even exceeds, that of some coarse-grained NiTi or Cu-based SMAs in terms of transformation reversibility and retention of functional strain (J. Ma et al, Acta Mater. 58, 2216 (2010); R. Kainuma et al, Shap. Mem. Superelasticity 4, 428 (2018)). These results are promising and demonstrate that this alloy is a good candidate for use in cyclic elastocaloric applications. Nevertheless, further improvement in fatigue resistance would be desirable to ensure long-term stability in real-world device environments.

Post-cycling EBSD phase mapping (Fig. 3b) reveals the presence of retained martensite, contributing to the minor residual strain (~1.3% after 1600 cycles). Microstructural optimization strategies, such as composition tuning to reduce defect formation, improve phase compatibility, or enhance interface mobility, may further suppress stabilized martensite and improve fatigue resistance.

We have added a one-sentence comment regarding the fatigue behavior in the revised Main Text on Page 2 Line 42, as follows:

“Note that the superelasticity remains functionally stable with increasing loading–unloading cycles. Room-temperature cyclic tests reveal persistence of the hysteresis loop beyond 1000 cycles with only minor residual strain (see Supplementary Information).”

We have also added these results and interpretations to the revised Supplementary Information, as follows:

“3. Cyclic mechanical tests at room temperature:

The mechanical compressive cycling behavior of the Ti–Al–Cr single crystal is shown in Supplementary Fig. 3. With increasing cycle number, the superelastic response evolves from an initially pronounced flag-shaped loop to a more linear stress–strain profile, resembling fatigue behavior commonly seen in shape memory alloys such as NiTi. The transformation stress hysteresis, measured at a fixed strain of 4%, remains nearly unchanged after mechanical training (Supplementary Fig. 3b). After 1600 cycles, the accumulated residual strain is limited to

approximately 1.3%, indicating modest functional degradation. Post-cycling EBSD analysis (Supplementary Fig. 3b) reveals the coexistence of B19 martensite and B2 parent phases, confirming that transformation-related microstructural features are preserved even after repeated cycling. These findings suggest that, although minor residual strain and mechanical training effects are present, the stress-induced martensitic transformation remains largely reversible.

Overall, this cyclic stability, both in mechanical response and microstructure, is encouraging for elastocaloric applications, where materials must undergo repeated actuation with minimal performance loss. However, from the viewpoint of long-term device operation, further enhancement in fatigue resistance remains an important direction for future research. For instance, microstructural optimization through alloy design strategies such as composition tuning or orientation controlling to suppress defect formation, promoting better phase compatibility, and enhancing interface mobility, may help mitigate the accumulation of stabilized martensite and delay fatigue-induced degradation. These approaches could extend the operational lifetime of the material under cyclic loading–unloading, making it more competitive for practical elastocaloric refrigeration systems.”

Fig. R1 (corresponding to Supplementary Fig. 3) (a) Cyclic stress–strain curves of a mechanically trained Ti–Al–Cr single-crystal specimen compressed along a near-[001] orientation, showing the evolution of the superelastic response over 1600 loading–unloading cycles at room temperature. (b) Evolution of residual strain and stress hysteresis as a function of cycle number. The stress hysteresis is defined as the difference between the loading and unloading stresses at a fixed strain of 4%. The recoverable strain remains high (~4%), and the hysteresis width stabilizes after the initial mechanical training cycles. The inset shows the EBSD phase map obtained after cyclic testing, revealing the coexistence of the B2 parent phase and residual B19 martensite.

Reviewer #2 (Remarks to the Author):

Dear Editor, Dear Authors;

The manuscript presents an elastocaloric effect of novel shape memory Ti-Al-Cr alloy, which shows reversed superelasticity and elastocaloric effect in a very wide temperature range – from ultra-low towards room temperatures. This makes this alloy very interesting for cryogenic and room temperature applications. The paper is interesting and well written, but some comments should be addressed.

Response:

We sincerely thank the reviewer for the positive and encouraging evaluation of our work. In response to the reviewer's constructive comments, we have revised the manuscript accordingly and provided detailed point-by-point responses below.

1. The authors claim a very high cooling capacity, which was calculated by multiplying isothermal entropy change and an available temperature range. I am aware that this metric has been used in literature before, but as such does not have any physical meaning according to my knowledge. The actual (meaningful) specific cooling capacity which elastocaloric material could produce should be calculated as a product of specific heat and the adiabatic temperature change or as a product of isothermal entropy change and a temperature (at which the cooling power is absorbed and not the entire possible temperature range).

Response:

We thank the reviewer for this valuable comment regarding the physical interpretation of cooling capacity. In our manuscript, we employed the definition of cooling capacity as the integral of the isothermal entropy change (ΔS) over a usable temperature span, i.e., $\int \Delta S dT$. This approach, while not directly representing the energy extracted in a single thermodynamic cycle, was originally applied for magnetocaloric materials whose operational temperature range is usually narrow. By integrating the entropy change over temperature, this metric provides a thermodynamic upper bound of the latent heat that the material can potentially offer across a given range.

For elastocaloric materials, which exhibit wider superelastic windows and are compatible with strain-based thermal cascading, this definition gains additional relevance. In particular, it offers a practical way to estimate the refrigerant capacity in future multistage cooling architectures, where a single material may be deployed across multiple thermal modules operating at different temperature biases. In such designs, the total entropy change integrated over the entire working range becomes a meaningful measure of the cumulative thermal potential. Thus, although we fully agree that quantities like $c_p \cdot \Delta T_{ad}$ or $\Delta S \cdot T$ at the cooling point better reflect the instantaneous performance of a single-stage cooling cycle, we believe that the entropy-integrated cooling

capacity still plays a useful conceptual role in guiding the development of wide-range or cascaded elastocaloric cooling systems.

To address the reviewer's concern, we have added additional clarification to this total cooling capacity quantity to the revised manuscript on Page 2 Line 7:

“This quantity does not represent the practical refrigeration capacity of a single-stage cooling device, which is more appropriately evaluated as the integral of ΔS over a specific operating temperature at the cooling point, or equivalently as the product of the specific heat and the adiabatic temperature change. Instead, the total cooling capacity can serve as a descriptive thermodynamic metric to quantify the combined effect of a large entropy change and an unusually broad operational temperature window, which may guide the development of wide-range or cascaded elastocaloric cooling systems using a single material [12][16][17][18].”

2. Was that the same sample that was tested from 4.2 K to 400 K or was it a different sample for each temperature?

Response:

We thank the reviewer for this important question. Two separate samples were used for the superelasticity and elastocaloric effect measurements, respectively. Both samples were cut from the same large single crystal, subjected to the same aging treatment and mechanical training process, and shared the identical crystallographic orientation. This approach ensured a consistent material state and minimized sample-to-sample variability, enabling reliable comparison across temperature-dependent experiments.

To clarify this point, we have added the following information to the *Methods* section of the revised manuscript:

“After stabilization of the superelastic response through mechanical cycling at room temperature (see Supplementary Information), a series of temperature-dependent loading–unloading tests were performed on a near [001]-oriented alloy single crystal sample A across a wide temperature range from 4.2 K to 400 K. Based on the stress-strain behavior observed, the elastocaloric effect was then measured on a second single-crystal sample (sample B), which had the same orientation and underwent the same mechanical training process as sample A.”

3. What was the strain rate at isothermal and adiabatic conditions? How the authors know that the selected strain-rate fulfills adiabatic conditions? Can they report adiabatic stress-strain responses?

Response:

We thank the reviewer for these insightful questions. As described in the *Methods* section, the strain rate under isothermal conditions was set to $5 \times 10^{-4} \text{ s}^{-1}$, while the unloading strain rate under

nominally adiabatic conditions was 0.2 s^{-1} , approximately 400 times faster. This value corresponds to the maximum achievable rate of our mechanical testing system. Based on prior studies of elastocaloric materials, strain rates on the order of 0.1 s^{-1} are generally accepted as sufficient to approximate adiabatic conditions in laboratory-scale mechanical testing setups (e.g., S. Qian et al., APL 111, 223902 (2017)). Therefore, we consider the applied rate of 0.2 s^{-1} to be representative of near-adiabatic unloading conditions.

To directly assess the influence of strain rate, we conducted additional experiments and compared stress–strain behavior at room temperature under both quasi-static and dynamic conditions. As shown in **Fig. R2**, no significant differences were observed in terms of critical stresses, hysteresis width. This insensitive strain-rate response suggests that the stress–strain curves measured under quasi-static conditions are also representative of those under near-adiabatic conditions.

We have added these results to the revised Supplementary Information, as follows:

“4. Strain rate dependence of superelastic response

We investigated the strain-rate dependence of the superelastic response in the Ti–Al–Cr single crystal at room temperature. Supplementary Fig. 4 shows the stress–strain curves measured under compressive loading–unloading at three representative strain rates: 0.0005 s^{-1} , 0.05 s^{-1} , and 0.2 s^{-1} . The specimen used was a mechanically trained near-[001]-oriented single crystal, identical in orientation and thermal–mechanical history to those used in elastocaloric experiments. As shown in Supplementary Fig. 4, all three stress–strain curves exhibit similar characteristics, including transformation start/finish stresses, stress hysteresis, and recoverable strain values. No significant shift in critical stresses or degradation of transformation behavior was observed across more than two orders of magnitude in strain rate.

Fig. R2 (corresponding to Supplementary Fig. 4) Mechanical response of another mechanically trained near-[001]-oriented Ti–Al–Cr single-crystal specimen under compressive loading–unloading at various strain rates at room temperature.

4. On Fig. 3c the authors are comparing adiabatic temperature changes calculated/measured by different methods, which is very interesting. However, they include both temperature-induced (calorimetry) and stress-induced. The author should comment on that and explain the reasons for deviations more in detail.

Response:

We thank the reviewer for this insightful comment. In Fig. 3c of the main text, we compare the adiabatic temperature change (ΔT_{ad}) of our Ti–Al–Cr alloy estimated via several approaches. However, the figure legend may have caused confusion: it does not compare different types of ΔT_{ad} directly, but rather visualizes the connection between different entropy change (ΔS) estimation method, namely, (i) calorimetric analysis, and (ii) Clausius–Clapeyron (C–C) relation-based derivation from stress–strain curves, and their corresponding implications for ΔT_{ad} (either calculated or measured). We now clarify this in the revised legend. Here, we provide further clarification to avoid potential confusion.

Specifically, the ΔS derived via calorimetry was not derived from a thermally induced martensitic transformation, since no such thermal transition occurs in the current Ti–Al–Cr alloy, but rather from independent specific heat measurements of each phase using the relaxation method. As such, it reflects the equilibrium thermodynamic difference between the two phases and can be regarded as a reliable estimate of the intrinsic transformation entropy. By contrast, ΔS derived via the C–C relation assumes a purely thermodynamic transformation path during stress-induced loading and neglects lattice dynamical contributions. However, in the present Ti–Al–Cr alloy, the martensitic transformation is additionally influenced by lattice dynamical constraints, as previously reported in our study (Ref. 25). This makes the C–C-derived ΔS intrinsically underestimated, whereas the calorimetric ΔS provides a more realistic upper bound for calculating ΔT_{ad} .

The directly measured ΔT_{ad} values, obtained via a thermocouple during fast unloading, are lower than those calculated using the calorimetric ΔS . This deviation is attributed to two key experimental limitations: (i) non-ideal adiabaticity during unloading, and (ii) the finite response speed and thermal resistance of the contact-type thermocouple, both of which result in underestimation of the true peak ΔT_{ad} . Additionally, incomplete transformation further reduces the observable ΔT_{ad} in practice.

We have added the following clarification on Page 4 Line 36:

“This entropy change was derived from calorimetric measurements of separately stabilized parent

and martensite phases and therefore reflects a stress-induced, rather than thermally induced, martensitic transformation.”

We have also added the following clarification on Page 5 Line 13:

“Deviations between calculated and measured adiabatic temperature changes primarily arise from imperfect adiabatic conditions during mechanical testing, sensor-related heat losses, and possible incomplete phase transformation under uniaxial compressive stresses.”

We have also revised the legend of fig.3c as follows to increase readability:

“Using calorimetry-derived ΔS ; Directly measured; Using C–C-derived ΔS .”

We have also added explanations to the revised Supplementary Information as follows:

“9. Comparison of entropy change (ΔS) derivation methods

In this work, the entropy change associated with the stress-induced martensitic transformation in the Ti–Al–Cr alloy is evaluated using three different approaches, each relying on distinct physical assumptions and experimental conditions. A comparison of these methods is summarized below to clarify their respective scope, advantages, and limitations.

(i) Calorimetric method based on heat capacity measurements

The entropy change is obtained by integrating the difference in specific heat between the B2 parent phase and the B19 martensite phase, which were independently measured using the relaxation calorimetry method. Although no thermally induced martensitic transformation occurs in this alloy, this approach directly reflects the equilibrium thermodynamic entropy difference between the two phases. As such, it is largely free from mechanical dissipation effects and kinetic constraints. This method is therefore considered to provide the most reliable estimate of the intrinsic transformation entropy and represents an upper bound for the achievable elastocaloric temperature change under ideal adiabatic and fully reversible conditions. The effective entropy change for elastocaloric applications should further consider dissipative contributions, including hysteresis loss.

(ii) Clausius–Clapeyron (C–C) relation derived from stress–strain behavior

The entropy change can also be estimated from the temperature dependence of the transformation stress using the Clausius–Clapeyron relation, assuming thermodynamic equilibrium during the stress-induced transformation. This approach is widely used in elastocaloric materials due to its experimental convenience and its direct link to mechanical measurements. In many conventional shape memory alloys, the entropy changes derived from calorimetry and the C–C relation is comparable. However, in the present Ti–Al–Cr alloy, the temperature dependence of the transformation stress is strongly influenced by anomalous elastic hardening and lattice dynamical effects. As a result, the assumptions underlying the C–C relation are not strictly satisfied, leading to an underestimation of ΔS when this method is applied. This

highlights a limitation of the C–C approach in systems where transformation kinetics and lattice dynamics play a significant role beyond classical thermodynamics.

(iii) Back-calculation from experimentally measured ΔT_{ad}

An effective entropy change can be inferred from the directly measured adiabatic temperature change using the relation $\Delta S_{eff} = -C_p \Delta T_{ad}/T$. This method reflects the actual entropy exchange occurring during a mechanical cycle under realistic experimental conditions. It inherently includes the effects of non-ideal adiabaticity, thermal losses, incomplete transformation, and hysteresis-related dissipation. Consequently, the resulting ΔS represents a lower bound of the transformation entropy and is particularly relevant for assessing practical device-level performance. Its limitation lies in its sensitivity to experimental conditions and measurement techniques, such as strain rate and temperature sensing accuracy.”

5. I would suggest that the authors calculate the material COP: mass x specific_heat adiabatic_temp_change vs adiabatic hysteresis input work.

Response:

Thank you for this valuable suggestion. Following the reviewer’s suggestion, we have calculated the material coefficient of performance (COP_{mater}) at room temperature. The COP was determined as the ratio between the absorbed heat during adiabatic cooling and the mechanical work input associated with hysteresis, using the specific heat capacity of the parent phase ($588 \text{ J}\cdot\text{kg}^{-1}\cdot\text{K}^{-1}$), the measured adiabatic temperature change (9.8 K), and the hysteretic energy loss derived from the whole stress–strain loop ($1260 \text{ J}\cdot\text{kg}^{-1}$). The COP_{mater} is calculated to be 4.6, comparable to that of commercial coarse-grained Ni–Ti elastocaloric materials.

However, in practical elastocaloric applications such as the single-stage elastocaloric testing system, COP_{mater} alone is not a sufficient metric for comparison, since elastocaloric refrigerants operate between a low-temperature heat source (T_c) and high temperature heat sink (T_h). Therefore, the coefficient of performance based on the reverse Stirling cycle ($COP_{stirling}$) was also calculated to provide a more realistic performance metric for comparing different elastocaloric materials. The elastocaloric Stirling-like cycle consists of isothermal loading and unloading processes, and two heat transfer processes under constant stress fields. And we have added this as a supplementary discussion.

We have added the following statement to the revised main text on Page 3 Line 38:

“At room temperature (294 K), the material coefficient of performance (COP_{mater}), defined as the ratio of the absorbed heat per unit mass to the input work per unit mass (i.e., the hysteretic mechanical work input), is calculated to be 4.6, comparable to that of commercial cocarse-grained Ni–Ti elastocaloric materials ^{[7][27]}.”

Moreover, we have also added Supplementary Table 1 and the corresponding discussion in the Supplementary Information to provide a comprehensive comparison of COP_{mater} , COP_{stirling} , and $COP_{\text{stirling}} / COP_{\text{carnot}}$ for various typical elastocaloric materials:

“10. Comparison of COP and efficiency among typical elastocaloric materials

A common metric used to evaluate the performance of elastocaloric materials is the material coefficient of performance (COP_{mater}), defined as the ratio of cooling output (Q) to the mechanical input work (W), assuming fully recoverable unloading energy and no auxiliary power consumption. Here, the cooling output Q is estimated from the measured adiabatic temperature change using $Q \approx C_p |\Delta T_{\text{ad}}|$, while the mechanical input work W is taken as the dissipated energy (denoted as Q_{diss} in the main text), evaluated from the hysteresis area of the stress–strain loop. However, in practical elastocaloric applications, COP_{mater} alone is not a sufficient metric for comparison, since elastocaloric refrigerants operate between a low-temperature heat source (T_c) and high-temperature heat sink (T_h).

Therefore, the coefficient of performance based on the reverse Stirling cycle (COP_{stirling}) is also calculated to provide a more realistic performance metric for comparing different elastocaloric materials operating under a reverse Stirling cycle^[4]:

$$COP_{\text{stirling}} = \frac{Q}{(T_h - T_c)\Delta S + W}. \quad (3)$$

Here, ΔS represents the isothermal entropy change associated with the martensitic transformation, which is assumed to remain approximately constant in the vicinity of ambient temperature. It should be noted that T_c and T_h may vary depending on device design and practical environments. Nevertheless, for a consistent comparison among different shape memory alloys, $T_c = 283 \text{ K}$ and $T_h = 293 \text{ K}$ are adopted following the reported literature^[4].

Under these conditions, the Carnot coefficient of performance for a heat pump operating between T_c and T_h , defined as:

$$COP_{\text{carnot}} = \frac{T_c}{T_h - T_c}, \quad (4)$$

is calculated to be 28.3. The normalized efficiency relative to the Carnot limit, expressed as the ratio of COP_{stirling} to COP_{carnot} , is then evaluated.

The summarized COP or efficiency values of the Ti–Al–Cr alloy and various shape memory alloys are listed in Supplementary Table 1. Although the COP_{stirling} of the present Ti–Al–Cr alloy is comparable to those of commercial NiTi and Cu–Zn–Al shape memory alloys, the $COP_{\text{stirling}} / COP_{\text{carnot}}$ ratio remains relatively lower than that of nanocrystalline NiTi or TiNiCuCo shape memory alloys. Further improvements, such as composition optimization to increase the isothermal entropy change or reduce hysteretic losses, may enhance the performance of the Ti–Al–Cr alloy for practical applications. It should also be noted that the present comparison is

limited to a narrow temperature span near room temperature and therefore does not fully reflect the cryogenic elastocaloric performance of the Ti–Al–Cr alloy.”

Supplementary Table 1. Comparison of cooling output, mechanical work, and COP values of representative elastocaloric materials under identical temperature span ($T_c = 283\text{ K}$, $T_h = 293\text{ K}$)

Material	$ Q $ (J g ⁻¹)	$ W $ (J g ⁻¹)	COP _{mater}	COP _{stirling}	COP _{carnot}	COP _{stirling} / COP _{carnot}	Ref.
NiTi	9.36	1.78	5.3	4.4	28.3	0.157	[5]
CuZnAl	6.05	1.00	6.1	5.0	28.3	0.176	[6]
NiMnTi	12.74	1.64	7.7	6.1	28.3	0.214	[7]
Nanocrystalline NiTi	16.3	1.34	12.1	8.5	28.3	0.300	[8]
TiNiCuCo	6.43	0.46	14.3	9.5	28.3	0.336	[9]
TiAlCr	5.76	1.26	4.6	3.9	28.3	0.138	This work

Reviewer #3 (Remarks to the Author):

In this manuscript (NCOMMS-25-83317-T), Song. et.al., reported an exceptionally large elastocaloric effect, in Ti-Al-Cr superelastic alloy, over an ultra-wide temperature range of 305 K with a cooling capacity reaching 4726 J/kg that is beyond Clausius-Clapeyron limit. They further explained their experimental observation originates from reversible stress-induced martensitic transition. In situ neutron diffraction measurements have been used to evaluate structural transition, during compression at room temperature. The manuscript is well written, but the following concerns must be addressed before considering in the “Nature Communications” journal.

Response: Thank you for your objective and in-depth comments on our work, and for offering such constructive suggestions. We have revised the manuscript accordingly to enhance the potential impact of our work. We are addressing the issue you raised by responding to your specific comments in detail in the following.

1. The authors reported in their previous paper {Ref.22} {stress vs temperatures (Fig.3(a)), Supply Fig.9 (a)-9(b)} are similar to the Fig. 1(a) and (b) of the present manuscript. They claimed an ultra-wide temperature window, while they have already reported on their earlier work based on the Figures of Ref: 22}. What is the novelty of the present work? Authors need to explain in detail.

Response:

We thank the reviewer for this important comment regarding the relationship between the present manuscript and our previous work (Ref. 25). Although both studies involve the same Ti–Al–Cr alloy system, the scientific focus, methodology, and central conclusions are fundamentally different.

First, Ref. 25 investigated tensile superelasticity primarily at cryogenic temperatures and reported an anomalous hardening behavior upon cooling. That work was focused on mechanical response and demonstrated the unusual temperature dependence of the critical stress. However, it did not systematically evaluate elastocaloric performance, entropy change, cooling capacity, or thermodynamic efficiency. No quantitative thermodynamic framework was established. In contrast, the present work shifts the focus from mechanical anomaly to thermodynamic functionality. It systematically investigates compressive superelasticity and directly measures the elastocaloric response (ΔT_{ad}), which is essential for realistic cooling applications where compressive loading is preferred for fatigue durability and device integration (e.g., Qian et al., Science 380, 722-727 (2023)). Systematic compressive elastocaloric behavior and cooling capacity were not examined in Ref. 25.

More importantly, the present study establishes a quantitative thermodynamic framework by

cross-validating entropy change through (i) Clausius–Clapeyron analysis, (ii) heat-capacity-based thermodynamic estimation, and (iii) direct ΔT_{ad} measurements. This integrated approach leads to a fundamentally new conclusion: the elastocaloric cooling temperature span and cooling capacity significantly exceed those predicted by conventional Clausius–Clapeyron scaling. This departure from classical thermodynamic expectations was not identified or analyzed in Ref. 25.

Therefore, the novelty of the present work lies not in re-reporting a wide temperature window, but in revealing its thermodynamic origin, demonstrating its functional consequence for elastocaloric cooling, and establishing a new design perspective in which stress–temperature coupling can be engineered beyond conventional Clausius–Clapeyron constraints.

We have added the following sentence in the Introduction section to the revised manuscript on Page 2 Line 28:

“Here, we report a new thermodynamic regime in a Ti–Al–Cr alloy, where the elastocaloric performance under compression exceeds Clausius–Clapeyron predictions, building upon but going significantly beyond our previous work.”

We have also added the following paragraph in the Discussion section to the revised manuscript on Page 6 Line 10:

“While our previous report established the foundational discovery of cryogenic superelasticity under tensile loading in this alloy system, the present study substantially broadens the scope by demonstrating that similar functionality can be achieved under compressive loading ^[25]. This finding is particularly important because compressive configurations are more compatible with the requirements of practical elastocaloric cooling devices ^{[12][26]}. Moreover, we performed a detailed and quantitative comparison between the experimentally measured elastocaloric response and predictions derived from several thermodynamic approaches. This integrated framework uncovers a fundamental deviation with the conventional Clausius–Clapeyron relation, revealing that both the operating temperature range and the cooling capacity significantly exceed theoretical expectations.”

2. It is stated in the first paragraph of page 2 that this material exhibits reversible stress-induced martensitic transformations from 4.2 K up to ~400 K in tension. How do they confirm that martensitic transformation (MT) (from Cubic B2 to orthorhombic B19 transition) is reversible, as martensitic transition is always associated with strain and hysteresis? Because hysteresis leads to work loss in every heat pumping cycle as dissipated heat. Henceforth, the authors are instructed to provide compatibility relations along with the cofactor conditions with the austenite and martensite interface. It is also suggested to show microstructure morphology during transition. See the reference; Cui et. al., Nature Materials 5, 286–290 (2006).

Response:

We appreciate the reviewer's critical comment regarding the reversibility of the martensitic transformation. In this study, we define reversibility based on the full strain recovery after unloading (Fig. 1a), stable functional response over multiple superelastic cycles (Supplementary Fig. 3a), and the reproducibility of adiabatic temperature change during repeated loading–unloading (Fig. 1c). These features collectively demonstrate excellent mechanical and functional reversibility over a broad temperature range. We agree that hysteresis implies some energy loss during transformation cycles; however, reversibility in the context of elastocaloric applications is commonly defined not by the absence of hysteresis, but by full strain recovery and long-term cyclic stability. Many commercial elastocaloric materials (e.g., Ni–Ti alloys) exhibit finite hysteresis while still delivering reversible transformation suitable for heat pumping applications.

Regarding microstructural evidence, in situ optical observations during tensile loading–unloading of a Ti–Al–Cr single crystal have been reported in our previous work (Ref. 25, *Nature* 2025), where nucleation, propagation, and complete disappearance of stress-induced martensite upon unloading were directly visualized. The present alloy exhibits the same B2 → B19 transformation crystallography and thermoelastic characteristics. Although high-resolution in situ microstructural characterization under the combined cryogenic and compressive-stress conditions of this study was not performed, the observed cyclic stability and full strain recovery are consistent with reversible interface motion.

To further address crystallographic compatibility, we evaluated the transformation stretch matrix U using lattice parameters obtained from neutron diffraction (Fig. 2). The calculated middle eigenvalue is $\lambda_2 = 1.01674$. While this value does not strictly satisfy the exact compatibility condition ($\lambda_2 = 1$), previous theoretical and experimental analyses (e.g. Cui et. al., *Nature Materials* 5, 286–290 (2006)) have shown that λ_2 approaching unity corresponds to improved lattice compatibility and reduced elastic strain at the austenite–martensite interface. Therefore, the near-unity λ_2 obtained here indicates favorable crystallographic compatibility, consistent with the experimentally observed reversible superelastic response.

3. It is requested to perform DSC measurement to determine the magnitude of hysteresis during transformation.

Response:

Thank you for raising this point. We would like to emphasize that the Ti–Al–Cr alloy does not undergo a cooling-induced martensitic transformation. This has been conclusively demonstrated in our previous work using both electrical resistivity and neutron diffraction measurements (see Extended Data Fig. 4 and Extended Data Fig. 8, respectively, in Song, Y. *et al.*, *Nature* **638**, 965–

971 (2025)). Specifically, the electrical resistivity measurements show neither a hysteresis loop nor an abrupt change in resistivity during cooling and heating between 400 K and 4 K, indicating the absence of a thermally induced martensitic transformation. Consistently, neutron diffraction results reveal no structural phase transition upon cooling.

As a result, DSC is not suitable for determining the transformation hysteresis in this alloy system, since no thermally driven martensitic transformation occurs. Instead, the martensitic transformation in this Ti–Al–Cr alloy is exclusively stress-induced, and the associated hysteresis is therefore evaluated through mechanical loading–unloading experiments rather than thermal analysis.

4. In general, the microstructure of the caloric materials during transformation changes drastically in consecutive transformation cycles, whereas macroscopic properties such as transformation temperature and latent heat are nearly reproducible. It is also suggested to show microstructure morphology during transition. See the reference Song et.al., Nature 502, 85–88 (2013).

Response:

Thank you for this insightful question. We have investigated the evolution of the anomalous mechanical behavior under repeated mechanical cycling by conducting cyclic loading–unloading tests for up to 1600 cycles. As shown in Fig. R1a, the superelastic response gradually evolves from a pronounced flag-shaped loop to a more linear stress–strain behavior with increasing cycle number, resembling the fatigue behavior commonly observed in commercial Ni–Ti shape memory alloys.

The transformation hysteresis was evaluated at a fixed strain of 4% and shows no significant change after mechanical training, as illustrated in Fig. R1b. After 1600 cycles, a residual strain of approximately 1.3% is accumulated, indicating limited functional degradation under cyclic loading. To further examine the microstructural evolution associated with cycling, the post-cycling microstructure was characterized using EBSD phase mapping (Fig. R1b). As shown in Fig. R1b, the presence of the B19 martensite phase is clearly observed within the matrix of the parent B2 phase, confirming the retention of transformation-related microstructural features after repeated cycling.

These results indicate that, although some mechanical training effects and residual strain accumulation occur during cycling, the stress-induced martensitic transformation remains largely reversible. This cyclic stability of both mechanical response and microstructural state is a critical requirement for practical elastocaloric applications, in which materials are subjected to repeated actuation over many cycles.

To address this point, we have added Fig. R1 as well as related discussion to the Supplementary Information to illustrate the cyclic mechanical response, hysteresis evolution, and associated microstructural stability:

“3. Cyclic mechanical tests at room temperature:

The mechanical compressive cycling behavior of the Ti–Al–Cr single crystal is shown in Supplementary Fig. 3. With increasing cycle number, the superelastic response evolves from an initially pronounced flag-shaped loop to a more linear stress–strain profile, resembling fatigue behavior commonly seen in shape memory alloys such as NiTi. The transformation stress hysteresis, measured at a fixed strain of 4%, remains nearly unchanged after mechanical training (Supplementary Fig. 3b). After 1600 cycles, the accumulated residual strain is limited to approximately 1.3%, indicating modest functional degradation. Post-cycling EBSD analysis (Supplementary Fig. 3b) reveals the coexistence of B19 martensite and B2 parent phases, confirming that transformation-related microstructural features are preserved even after repeated cycling. These findings suggest that, although minor residual strain and mechanical training effects are present, the stress-induced martensitic transformation remains largely reversible.

Overall, this cyclic stability, both in mechanical response and microstructure, is encouraging for elastocaloric applications, where materials must undergo repeated actuation with minimal performance loss. However, from the viewpoint of long-term device operation, further enhancement in fatigue resistance remains an important direction for future research. For instance, microstructural optimization through alloy design strategies such as composition tuning or orientation controlling to suppress defect formation, promoting better phase compatibility, and enhancing interface mobility, may help mitigate the accumulation of stabilized martensite and delay fatigue-induced degradation. These approaches could extend the operational lifetime of the material under cyclic loading–unloading, making it more competitive for practical elastocaloric refrigeration systems.”

Fig. R1 (corresponding to Supplementary Fig. 3) (a) Cyclic stress–strain curves of a

mechanically trained Ti–Al–Cr single-crystal specimen compressed along a near-[001] orientation, showing the evolution of the superelastic response over 1600 loading–unloading cycles at room temperature. (b) Evolution of residual strain and stress hysteresis as a function of cycle number. The stress hysteresis is defined as the difference between the loading and unloading stresses at a fixed strain of 4%. The recoverable strain remains high (~4%), and the hysteresis width stabilizes after the initial mechanical training cycles. The inset shows the EBSD phase map obtained after cyclic testing, revealing the coexistence of the B2 parent phase and residual B19 martensite.

5. The author should add material's efficiency of the present sample i.e., material's coefficient of performance (COP) to Carnot efficiency (COP) and a comparison figure for the material with other elastocaloric materials.

Response:

Thank you for this valuable suggestion. Following the reviewer's suggestion, we have calculated the material coefficient of performance (COP_{mater}) at room temperature. The COP was determined as the ratio between the absorbed heat during adiabatic cooling and the mechanical work input associated with hysteresis, using the specific heat capacity of the parent phase ($588 \text{ J}\cdot\text{kg}^{-1}\cdot\text{K}^{-1}$), the measured adiabatic temperature change (9.8 K), and the hysteretic energy loss derived from the whole stress–strain loop ($1260 \text{ J}\cdot\text{kg}^{-1}$). The COP_{mater} is calculated to be 4.6, comparable to that of commercial coarse-grained Ni–Ti elastocaloric materials.

However, in practical elastocaloric applications such as the single-stage elastocaloric testing system, COP_{mater} alone is not a sufficient metric for comparison, since elastocaloric refrigerants operate between a low-temperature heat source (T_c) and high temperature heat sink (T_h). Therefore, the coefficient of performance based on the reverse Stirling cycle (COP_{stirling}) was also calculated to provide a more realistic performance metric for comparing different elastocaloric materials. The elastocaloric Stirling-like cycle consists of isothermal loading and unloading processes, and two heat transfer processes under constant stress fields. And we have added this as a supplementary discussion.

We have added the following statement to the revised main text on Page 3 Line 38:

“At room temperature (294 K), the material coefficient of performance (COP_{mater}), defined as the ratio of the absorbed heat per unit mass to the input work per unit mass (i.e., the hysteretic mechanical work input), is calculated to be 4.6, comparable to that of commercial coarse-grained Ni–Ti elastocaloric materials.”

Moreover, we have also added Supplementary Table 1 and the corresponding discussion in the Supplementary Information to provide a comprehensive comparison of COP_{mater} , COP_{stirling} , and

$COP_{\text{stirling}} / COP_{\text{carnot}}$ for various typical elastocaloric materials:

“10. Comparison of COP and efficiency among typical elastocaloric materials

A common metric used to evaluate the performance of elastocaloric materials is the material coefficient of performance (COP_{mater}), defined as the ratio of cooling output (Q) to the mechanical input work (W), assuming fully recoverable unloading energy and no auxiliary power consumption. Here, the cooling output Q is estimated from the measured adiabatic temperature change using $Q \approx C_p |\Delta T_{ad}|$, while the mechanical input work W is taken as the dissipated energy (denoted as Q_{diss} in the main text), evaluated from the hysteresis area of the stress–strain loop. However, in practical elastocaloric applications, COP_{mater} alone is not a sufficient metric for comparison, since elastocaloric refrigerants operate between a low-temperature heat source (T_c) and high-temperature heat sink (T_h).

Therefore, the coefficient of performance based on the reverse Stirling cycle (COP_{stirling}) is also calculated to provide a more realistic performance metric for comparing different elastocaloric materials operating under a reverse Stirling cycle^[4]:

$$COP_{\text{stirling}} = \frac{Q}{(T_h - T_c)\Delta S + W} \quad (3)$$

Here, ΔS represents the isothermal entropy change associated with the martensitic transformation, which is assumed to remain approximately constant in the vicinity of ambient temperature. It should be noted that T_c and T_h may vary depending on device design and practical environments. Nevertheless, for a consistent comparison among different shape memory alloys, $T_c = 283$ K and $T_h = 293$ K are adopted following the reported literature [4].

Under these conditions, the Carnot coefficient of performance for a heat pump operating between T_c and T_h , defined as:

$$COP_{\text{carnot}} = \frac{T_c}{T_h - T_c} \quad (4)$$

is calculated to be 28.3. The normalized efficiency relative to the Carnot limit, expressed as the ratio of COP_{stirling} to COP_{carnot} , is then evaluated.

The summarized COP or efficiency values of the Ti–Al–Cr alloy and various shape memory alloys are listed in Supplementary Table 1. Although the COP_{stirling} of the present Ti–Al–Cr alloy is comparable to those of commercial NiTi and Cu–Zn–Al shape memory alloys, the $COP_{\text{stirling}} / COP_{\text{carnot}}$ ratio remains relatively lower than that of nanocrystalline NiTi or TiNiCuCo shape memory alloys. Further improvements, such as composition optimization to increase the isothermal entropy change or reduce hysteretic losses, may enhance the performance of the Ti–Al–Cr alloy for practical applications. It should also be noted that the present comparison is limited to a narrow temperature span near room temperature and therefore does not fully reflect the cryogenic elastocaloric performance of the Ti–Al–Cr alloy.”

Supplementary Table 1. Comparison of cooling output, mechanical work, and COP values of representative elastocaloric materials under identical temperature span ($T_c = 283$ K, $T_h = 293$ K)

Material	$ Q $ (J g ⁻¹)	$ W $ (J g ⁻¹)	COP _{mater}	COP _{stirling}	COP _{carnot}	COP _{stirling} / COP _{carnot}	Ref.
NiTi	9.36	1.78	5.3	4.4	28.3	0.157	[5]
CuZnAl	6.05	1.00	6.1	5.0	28.3	0.176	[6]
NiMnTi	12.74	1.64	7.7	6.1	28.3	0.214	[7]
Nanocrystalline NiTi	16.3	1.34	12.1	8.5	28.3	0.300	[8]
TiNiCuCo	6.43	0.46	14.3	9.5	28.3	0.336	[9]
TiAlCr	5.76	1.26	4.6	3.9	28.3	0.138	This work

6. Have the authors checked their mechanical stability, such as bulk modulus B to shear modulus ratio and the Cauchy pressure of the material?

Response:

We thank the reviewer for this valuable suggestion. In the present study, our primary focus was on the experimental demonstration of cryogenic superelasticity and elastocaloric cooling performance. However, as the reviewer correctly points out, an assessment of mechanical stability is also important.

We have therefore evaluated the mechanical stability of the B2 parent phase based on previously reported elastic constants. At room temperature, the elastic constants are $C_{11}=129.5$ GPa, $C_{12}=101.8$ GPa, and $C_{44}=62.8$ GPa. These values satisfy the Born mechanical stability criteria ($C_{11}-C_{12}>0$, $C_{44}>0$, and $C_{11}+2C_{12}>0$) for cubic crystals, confirming the elastic stability of the B2 phase.

The bulk modulus is calculated as $B = (C_{11} + 2C_{12})/3 \approx 111$ GPa. Using the Voigt–Reuss–Hill averaged shear modulus ($G \approx 35$ GPa), the B/G ratio is approximately 3.2, which exceeds the empirical ductility criterion of 1.75, indicating a ductile character of the B2 phase. It should be noted that while the B/G ratio is an empirical indicator primarily developed for isotropic polycrystalline materials, it nevertheless provides a useful qualitative measure of ductility in the present cubic system. The Cauchy pressure ($C_{11}-C_{12} \approx 39$ GPa) is positive, consistent with metallic bonding and mechanical robustness.

These results support the mechanical stability of the B2 matrix and its ability to sustain repeated stress-induced martensitic transformation under superelastic loading.

A brief discussion of the B/G ratio and Cauchy pressure has been added to the Supplementary Information to address this point:

“ Meanwhile, the Cauchy pressure and the bulk-modulus-to-shear-modulus ratio for the cubic crystal can be evaluated from the elastic constants. At room temperature, $C_{11} = 129.5$ GPa, $C_{12} = 101.8$ GPa, and $C_{44} = 62.8$ GPa^[1], giving a Cauchy pressure ($C_{12} - C_{44}$) of approximately 39 GPa, characteristic of metallic bonding and consistent with ductile behavior. The bulk modulus is $B \approx 111$ GPa. Using the Voigt–Reuss–Hill averaged shear modulus ($G \approx 35$ GPa), the B/G ratio is approximately 3.2, exceeding the empirical ductility threshold of 1.75.

It is worth noting that cubic single crystals exhibit elastic anisotropy, and the shear modulus depends on deformation mode. In particular, the tetragonal shear modulus $C' = (C_{11} - C_{12})/2 \approx 13.9$ GPa is relatively small, and the elastic anisotropy factor $A = C_{44}/C' \approx 4.5$, reflecting a pronounced soft shear mode associated with lattice instability and facilitating stress-induced martensitic transformation. This elastic softness is consistent with the observed superelastic response.”

7. They performed the orientation dependence of recoverable strain under compression, which has depicted in Fig.2(f). In addition, a change of grain boundaries can enhance strain recoverability and elevate the utilization of latent heat during stress-driven MT. While improving strength by reducing grain size is an established physical principle, i.e., the Hall-Petch equation. I think it would be more persuasive and powerful if the author could comment on this and explain how the grain size of their sample can help improve strain recoverability?

Response:

We thank the reviewer for this insightful comment. Grain refinement is indeed an effective strategy for strengthening conventional metallic materials through the Hall–Petch mechanism. However, the role of grain size in shape memory alloys is fundamentally different from its role in conventional plastic strengthening.

In stress-driven martensitic transformation, strain compatibility across grain boundaries becomes a dominant factor. In the present Ti–Al–Cr alloy, excellent superelasticity is achieved in single-crystal form. In contrast, polycrystalline samples exhibit significantly degraded recoverable strain due to strain incompatibility at grain boundaries and triple junctions during transformation. These interfaces act as stress concentrators and promote premature cracking, thereby limiting recoverable strain. Therefore, unlike conventional Hall–Petch strengthening, reducing grain size does not necessarily enhance recoverable strain in this alloy system. Instead, transformation compatibility and interfacial stability govern superelastic performance. As reported for copper-based shape memory alloys (Sutou et al., *Acta Mater* 53, 4121-4133 (2005); Ueland et al., *Adv. Funct. Mater.* 22, 2094-2099 (2012).), superior recoverability can be achieved when grain boundary constraints are minimized. The present alloy follows a similar behavior, where single-crystal or boundary-minimized configurations are more favorable for achieving large reversible

strain.

We have added the following comment to the revised main text on Page 3 Line 20:

“Additionally, it is worth noting that in stress-driven martensitic transformation, strain compatibility across grain boundaries becomes a dominant factor, and intergranular constraint can limit reversible strain ^[27] ^[28]. Therefore, configurations with reduced grain-boundary constraint, such as single-crystal specimens, are more favorable for achieving large superelastic strain in the present alloy system.”

8. In the Summary section, the authors compared β -phase Ti alloy with their findings. Recently, a colossal elastocaloric effect has been observed in a promising all-d-Heusler Ni-Mn-Ti ferroelastic system. Authors should add the ferroelastic Ni-Mn-Ti material in Fig. 4(a) and explain their findings with these systems. See the reference: [Cong. et. al., Physical Review Letters 122, 255703 (2019)].

Response:

Thank you for this valuable suggestion. We fully acknowledge that the all-d-Heusler Ni-Mn-Ti ferroelastic system holds the current record for adiabatic temperature change during reversible martensitic transformation. As reported by Cong et al. (Phys. Rev. Lett. 122, 255703 (2019)), the exceptionally large ΔT_{ad} originates from a strong coupling between transformation entropy and the relative unit-cell volume change ($\Delta V/V_0$), further enhanced by magneto-structural interactions.

Following the reviewer’s advice, we have instead included the Ni-Mn-Ti ferroelastic alloy in Fig. 4(b), where elastocaloric materials across different alloy systems are compared. This placement allows a more appropriate cross-system comparison while preserving the focus of Fig. 4(a).

In addition, we have added the following clarification to the main text to explicitly address the relationship between our findings and the Ni-Mn-Ti system to the main text (Page 6 Line 28):

“To date, the largest adiabatic temperature change has been reported in the Ni-Mn-Ti system near room temperature owing to the strong coupling among martensitic transformation entropy, unit-cell volume change, and magnetic interactions, whereas the present Ti-Al-Cr alloy offers a much wider operating temperature range while maintaining a moderately large adiabatic temperature change ^[23].”

RESPONSE TO REVIEWERS' COMMENTS

Reviewer #1 (Remarks to the Author):

The Authors properly answered to all the issues and queries. Moreover, they added substantial observations in the text and in some cases they provided additional tests to improve their discussion. Therefore, the quality of the manuscript is substantially improved and the relevance of the work is confirmed. In my opinion, the manuscript is now suitable for publication in the present form.

Response: We sincerely thank the reviewer for the positive evaluation and for the valuable time and effort devoted to assessing our manuscript. We greatly appreciate your constructive comments and suggestions throughout the review process, which have significantly improved the quality and clarity of our work.

Reviewer #2 (Remarks to the Author):

The authors have successfully addressed most of my previous comments, however, one important issue remains. Although the authors agreed that the reported total cooling capacity does not represent the practical refrigeration capacity of a cooling device (and is thus as such more or less meaningless), they have still reported this value (in abstract, which might be misleading!). Later in the text (within the COP calculation) they have calculated also more representative cooling capacity (specific heat multiplied by adiabatic temperature changes), and I would strongly suggest reporting this value instead.

Response: We greatly appreciate the reviewer's careful evaluation and constructive suggestions. We agree that the previously reported total cooling capacity may not accurately reflect the practical refrigeration performance of elastocaloric materials and could be misleading if presented without clarification. In response, we have revised the manuscript accordingly. Specifically, the total cooling capacity has been removed from the Abstract, and more application-relevant performance metrics, including the cooling output and COP_{mater} at room temperature, are now emphasized.

The revised Abstract now reads:

“The elastocaloric effect, driven by stress-induced martensitic transformations, offers a promising route toward efficient and environmentally friendly solid-state cooling. However, its practical implementation has been hindered by an inherent trade-off: materials exhibiting large isothermal entropy changes (ΔS) typically operate over narrow temperature windows, thereby

limiting their overall cooling performance. Here, we demonstrate an elastocaloric response in a Ti–Al–Cr superelastic alloy that overcomes this limitation. Direct measurements reveal a pronounced elastocaloric effect over an ultra-wide temperature range of 305 K, from 97 K to 402 K. This temperature span exceeds that predicted by the Clausius–Clapeyron relationship (235 K), indicating a significant deviation from conventional thermodynamic expectations. At room temperature, a large adiabatic temperature change of ~10 K is directly measured, corresponding to a cooling output of 5.76 J·g⁻¹ of a coefficient of performance (COP_{mater}) of 4.6, demonstrating competitive cooling performance at practical operating conditions. In addition, the elastocaloric response is maintained over the entire temperature range despite the expected decrease in entropy change at lower temperatures, indicating that the conventional trade-off between temperature span and cooling strength is effectively mitigated. This exceptional behavior originates from a combination of anomalous temperature dependence of the critical stress for martensitic transformation and high mechanical strength, which together enable fully reversible stress-induced transformations across a broad thermal domain. Our findings reveal a new regime of elastocaloric behavior and establish a guiding principle for overcoming the apparent limitations imposed by Clausius–Clapeyron-based descriptions in caloric materials.”

We have also added the cooling performance metrics at room temperature in the main text (page 3, line 43), as follows:

“At room temperature (294 K), the material coefficient of performance (COP_{mater}), defined as the ratio of the cooling output per unit mass to the input work per unit mass (i.e., the hysteretic mechanical work), is calculated to be 4.6, based on a cooling output of 5.76 J·g⁻¹ derived from a |ΔT_{ad}| of 9.8 K. This value is comparable to that of commercial coarse-grained Ni–Ti elastocaloric materials^{[7][29]}.”

Accordingly, we have added the calculation details in the Supplementary Information, as follows:

“Using the specific heat capacity of the parent phase (588 J·kg⁻¹·K⁻¹), the measured adiabatic temperature change (9.8 K), and the hysteretic energy loss derived from the whole stress–strain loop (1260 J·kg⁻¹), the COP_{mater} at room temperature (294 K) for the present Ti–Al–Cr alloy is calculated to be 4.6.”

RESPONSE TO REVIEWERS' COMMENTS

Reviewer #2 (Remarks to the Author):

Dear Authors;

The revised manuscript now indeed reports the specific output cooling energy, which is much more relevant. However, I would strongly suggest updating the "Thermodynamic analysis and predictions" by adding the equation based on which the specific output cooling energy was calculated and comment on the difference between it and total cooling capacity (eq. (1)), which is still mentioned as a central figure of merit for elastocaloric material.

Response: Thank you for recognizing our efforts to improve the manuscript and for this helpful suggestion. In our previous version, the equation for the specific cooling output was provided in the Supplementary Information. Following the reviewer's recommendation, we have revised the manuscript to clearly distinguish between the total cooling capacity and the specific cooling output.

Specifically, we have added the equation used to calculate the specific cooling output and included a detailed explanation in the "*Thermodynamic analysis and predictions*" section (page 6, line 19), as follows:

*"It is important to distinguish between the total cooling capacity ($Q_{cooling}$) and the specific cooling output (Q_s) used in evaluating COP_{mater} (see **Supplementary Information**), which is expressed as:*

$$Q_s = C_p |\Delta T_{ad}| \quad (6)$$

Here, Q_s describes the cooling energy produced per superelastic cycle under near-ambient conditions, whereas $Q_{cooling}$ represents the total cooling energy that can be obtained over a given temperature span (e.g., in a cascading cooling configuration). While both quantities are useful, Q_s provides a more common metric for evaluating elastocaloric performance in practical operation, where the temperature dependence of elastocaloric properties is often neglected."